# ATTENTION, LEARN TO SOLVE ROUTING PROBLEMS!

**Wouter Kool**
University of Amsterdam
ORTEC
w.w.m.kool@uva.nl

**Herke van Hoof**
University of Amsterdam
h.c.vanhoof@uva.nl

**Max Welling**
University of Amsterdam
CIFAR
m.welling@uva.nl

## ABSTRACT

The recently presented idea to learn heuristics for combinatorial optimization problems is promising as it can save costly development. However, to push this idea towards practical implementation, we need better models and better ways of training. We contribute in both directions: we propose a model based on attention layers with benefits over the Pointer Network and we show how to train this model using REINFORCE with a simple baseline based on a deterministic greedy rollout, which we find is more efficient than using a value function. We significantly improve over recent learned heuristics for the Travelling Salesman Problem (TSP), getting close to optimal results for problems up to 100 nodes. With the same hyperparameters, we learn strong heuristics for two variants of the Vehicle Routing Problem (VRP), the Orienteering Problem (OP) and (a stochastic variant of) the Prize Collecting TSP (PCTSP), outperforming a wide range of baselines and getting results close to highly optimized and specialized algorithms.

## 1 INTRODUCTION

Imagine yourself travelling to a scientific conference. The field is popular, and surely you do not want to miss out on anything. You have selected several posters you want to visit, and naturally you must return to the place where you are now: the coffee corner. In which order should you visit the posters, to minimize your time walking around? This is the Travelling Scientist Problem (TSP).

You realize that your problem is equivalent to the Travelling Salesman Problem (conveniently also TSP). This seems discouraging as you know the problem is (NP-)hard (Garey & Johnson, 1979). Fortunately, complexity theory analyzes the worst case, and your Bayesian view considers this unlikely. In particular, you have a strong prior: the posters will probably be laid out regularly. You want a special algorithm that solves not any, but *this* type of problem instance. You have some months left to prepare. As a machine learner, you wonder whether your algorithm can be learned?

**Motivation** Machine learning algorithms have replaced humans as the engineers of algorithms to solve various tasks. A decade ago, computer vision algorithms used hand-crafted features but today they are learned *end-to-end* by Deep Neural Networks (DNNs). DNNs have outperformed classic approaches in speech recognition, machine translation, image captioning and other problems, by learning from data (LeCun et al., 2015). While DNNs are mainly used to make *predictions*, Reinforcement Learning (RL) has enabled algorithms to learn to make *decisions*, either by interacting with an environment, e.g. to learn to play Atari games (Mnih et al., 2015), or by inducing knowledge through look-ahead search: this was used to master the game of Go (Silver et al., 2017).

The world is not a game, and we desire to train models that make decisions to solve real problems. These models must learn to select good solutions for a problem from a combinatorially large set of potential solutions. Classically, approaches to this problem of *combinatorial optimization* can be divided into *exact methods*, that guarantee finding optimal solutions, and *heuristics*, that trade off optimality for computational cost, although exact methods can use heuristics internally and vice versa. Heuristics are typically expressed in the form of rules, which can be interpreted as policies to make decisions. We believe that these policies can be parameterized using DNNs, and be trained to obtain new and stronger algorithms for many different combinatorial optimization problems, similar to the way DNNs have boosted performance in the applications mentioned before. In this paper, we focus on routing problems: an important class of practical combinatorial optimization problems.

The promising idea to learn heuristics has been tested on TSP (Bello et al., 2016). In order to push this idea, we need better models and better ways of training. Therefore, we propose to use a powerful model based on attention and we propose to train this model using REINFORCE with a simple but effective greedy rollout baseline. The goal of our method is not to outperform a non-learned, specialized TSP algorithm such as Concorde (Applegate et al., 2006). Rather, we show the flexibility of our approach on multiple (routing) problems of reasonable size, with *a single set of hyperparameters*. This is important progress towards the situation where we can learn strong heuristics to solve a wide range of different practical problems for which no good heuristics exist.

## 2 RELATED WORK

The application of Neural Networks (NNs) for optimizing decisions in combinatorial optimization problems dates back to Hopfield & Tank (1985), who applied a Hopfield-network for solving small TSP instances. NNs have been applied to many related problems (Smith, 1999), although in most cases in an *online* manner, starting 'from scratch' and 'learning' a solution for every instance. More recently, (D)NNs have also been used *offline* to learn about an entire class of problem instances.

Vinyals et al. (2015) introduce the Pointer Network (PN) as a model that uses attention to output a permutation of the input, and train this model offline to solve the (Euclidean) TSP, supervised by example solutions. Upon test time, their beam search procedure filters invalid tours. Bello et al. (2016) introduce an Actor-Critic algorithm to train the PN without supervised solutions. They consider each instance as a training sample and use the cost (tour length) of a sampled solution for an unbiased Monte-Carlo estimate of the policy gradient. They introduce extra model depth in the decoder by an additional *glimpse* (Vinyals et al., 2016) at the embeddings, masking nodes already visited. For small instances ($n = 20$), they get close to the results by Vinyals et al. (2015), they improve for $n = 50$ and additionally include results for $n = 100$. Nazari et al. (2018) replace the LSTM encoder of the PN by element-wise projections, such that the updated embeddings after state-changes can be effectively computed. They apply this model on the Vehicle Routing Problem (VRP) with split deliveries and a stochastic variant.

Dai et al. (2017) do not use a separate encoder and decoder, but a single model based on graph embeddings. They train the model to output the *order* in which nodes are *inserted* into a partial tour, using a helper function to insert at the best possible location. Their 1-step DQN (Mnih et al., 2015) training method trains the algorithm per step and incremental rewards provided to the agent at every step effectively encourage greedy behavior. As mentioned in their appendix, they use the negative of the reward, which combined with discounting encourages the agent to insert the farthest nodes first, which is known to be an effective heuristic (Rosenkrantz et al., 2009).

Nowak et al. (2017) train a Graph Neural Network in a supervised manner to directly output a tour as an adjacency matrix, which is converted into a feasible solution by a beam search. The model is non-autoregressive, so cannot condition its output on the partial tour and the authors report an optimality gap of 2.7% for $n = 20$, worse than autoregressive approaches mentioned in this section. Kaempfer & Wolf (2018) train a model based on the Transformer architecture (Vaswani et al., 2017) that outputs a fractional solution to the multiple TSP (mTSP). The result can be seen as a solution to the linear relaxation of the problem and they use a beam search to obtain a feasible integer solution.

Independently of our work, Deudon et al. (2018) presented a model for TSP using attention in the OR community. They show performance can improve using 2OPT local search, but do not show benefit of their model in direct comparison to the PN. We use a different decoder and improved training algorithm, both contributing to significantly improved results, *without* 2OPT and additionally show application to different problems. For a full discussion of the differences, we refer to Appendix B.4.

## 3 ATTENTION MODEL

We define the Attention Model in terms of the TSP. For other problems, the model is the same but the input, mask and decoder context need to be defined accordingly, which is discussed in the Appendix. We define a problem instance $s$ as a graph with $n$ nodes, where node $i \in \{1, \ldots, n\}$ is represented by features $\mathbf{x}_i$. For TSP, $\mathbf{x}_i$ is the coordinate of node $i$ and the graph is fully connected (with self-connections) but in general, the model can be considered a Graph Attention Network (Velickovic

et al., 2018) and take graph structure into account by a masking procedure (see Appendix A). We define a solution (tour) $\boldsymbol{\pi} = (\pi_1, \ldots, \pi_n)$ as a permutation of the nodes, so $\pi_t \in \{1, \ldots n\}$ and $\pi_t \neq \pi_{t'} \, \forall t \neq t'$. Our attention based encoder-decoder model defines a stochastic policy $p(\boldsymbol{\pi}|s)$ for selecting a solution $\boldsymbol{\pi}$ given a problem instance $s$. It is factorized and parameterized by $\boldsymbol{\theta}$ as

$$p_{\boldsymbol{\theta}}(\boldsymbol{\pi}|s) = \prod_{t=1}^{n} p_{\boldsymbol{\theta}}(\pi_t|s, \boldsymbol{\pi}_{1:t-1}). \tag{1}$$

The encoder produces embeddings of all input nodes. The decoder produces the sequence $\boldsymbol{\pi}$ of input nodes, one node at a time. It takes as input the encoder embeddings and a problem specific mask and context. For TSP, when a partial tour has been constructed, it cannot be changed and the remaining problem is to find a path from the last node, through all unvisited nodes, to the first node. The order and coordinates of other nodes already visited are irrelevant. To know the first and last node, the decoder context consists (next to the graph embedding) of embeddings of the first and last node. Similar to Bello et al. (2016), the decoder observes a mask to know which nodes have been visited.

## 3.1 ENCODER

The encoder that we use (Figure 1) is similar to the encoder used in the Transformer architecture by Vaswani et al. (2017), but we do not use positional encoding such that the resulting node embeddings are invariant to the input order. From the $d_{\mathrm{x}}$-dimensional input features $\mathbf{x}_i$ (for TSP $d_{\mathrm{x}} = 2$), the encoder computes initial $d_{\mathrm{h}}$-dimensional node embeddings $\mathbf{h}_i^{(0)}$ (we use $d_{\mathrm{h}} = 128$) through a learned linear projection with parameters $W^{\mathrm{x}}$ and $\mathbf{b}^{\mathrm{x}}$: $\mathbf{h}_i^{(0)} = W^{\mathrm{x}}\mathbf{x}_i + \mathbf{b}^{\mathrm{x}}$. The embeddings are updated using $N$ attention layers, each consisting of two sublayers. We denote with $\mathbf{h}_i^{(\ell)}$ the node

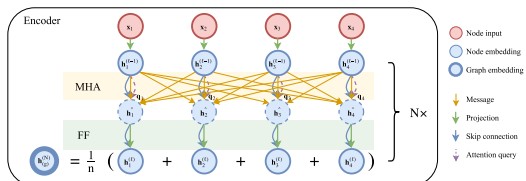

Figure 1: Attention based encoder. Input nodes are embedded and processed by $N$ sequential layers, each consisting of a multi-head attention (MHA) and node-wise feed-forward (FF) sublayer. The graph embedding is computed as the mean of node embeddings. Best viewed in color.

embeddings produced by layer $\ell \in \{1, .., N\}$. The encoder computes an aggregated embedding $\bar{\mathbf{h}}^{(N)}$ of the input graph as the mean of the final node embeddings $\mathbf{h}_i^{(N)}$: $\bar{\mathbf{h}}^{(N)} = \frac{1}{n}\sum_{i=1}^{n} \mathbf{h}_i^{(N)}$. Both the node embeddings $\mathbf{h}_i^{(N)}$ and the graph embedding $\bar{\mathbf{h}}^{(N)}$ are used as input to the decoder.

**Attention layer** Following the Transformer architecture (Vaswani et al., 2017), each attention layer consist of two sublayers: a multi-head attention (MHA) layer that executes message passing between the nodes and a node-wise fully connected feed-forward (FF) layer. Each sublayer adds a skip-connection (He et al., 2016) and batch normalization (BN) (Ioffe & Szegedy, 2015) (which we found to work better than layer normalization (Ba et al., 2016)):

$$\hat{\mathbf{h}}_i = \mathrm{BN}^{\ell}\left(\mathbf{h}_i^{(\ell-1)} + \mathrm{MHA}_i^{\ell}\left(\mathbf{h}_1^{(\ell-1)}, \ldots, \mathbf{h}_n^{(\ell-1)}\right)\right) \tag{2}$$

$$\mathbf{h}_i^{(\ell)} = \mathrm{BN}^{\ell}\left(\hat{\mathbf{h}}_i + \mathrm{FF}^{\ell}(\hat{\mathbf{h}}_i)\right). \tag{3}$$

The layer index $\ell$ indicates that the layers do *not* share parameters. The MHA sublayer uses $M = 8$ heads with dimensionality $\frac{d_h}{M} = 16$, and the FF sublayer has one hidden (sub)sublayer with dimension 512 and ReLu activation. See Appendix A for details.

## 3.2 DECODER

Decoding happens sequentially, and at timestep $t \in \{1, \ldots n\}$, the decoder outputs the node $\pi_t$ based on the embeddings from the encoder and the outputs $\pi_{t'}$ generated at time $t' < t$. During decoding, we augment the graph with a special *context node* ($c$) to represent the decoding context. The decoder computes an attention (sub)layer on top of the encoder, but with messages only to the context node for efficiency.[1] The final probabilities are computed using a single-head attention mechanism. See Figure 2 for an illustration of the decoding process.

---

[1] $n \times n$ attention between all nodes is expensive to compute in every step of the decoding process.

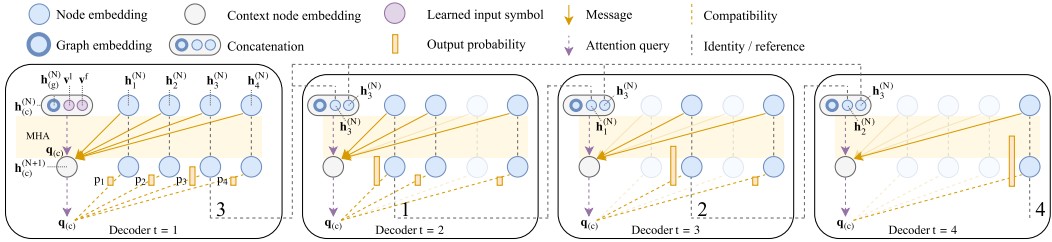

Figure 2: Attention based decoder for the TSP problem. The decoder takes as input the graph embedding and node embeddings. At each time step $t$, the context consist of the graph embedding and the embeddings of the first and last (previously output) node of the partial tour, where learned placeholders are used if $t = 1$. Nodes that cannot be visited (since they are already visited) are masked. The example shows how a tour $\boldsymbol{\pi} = (3, 1, 2, 4)$ is constructed. Best viewed in color.

**Context embedding**  The context of the decoder at time $t$ comes from the encoder and the output up to time $t$. As mentioned, for the TSP it consists of the embedding of the graph, the previous (last) node $\pi_{t-1}$ and the first node $\pi_1$. For $t = 1$ we use learned $d_{\mathrm{h}}$-dimensional parameters $\mathbf{v}^{\mathrm{l}}$ and $\mathbf{v}^{\mathrm{f}}$ as input placeholders:

$$\mathbf{h}_{(c)}^{(N)} = \begin{cases} \left[\bar{\mathbf{h}}^{(N)}, \mathbf{h}_{\pi_{t-1}}^{(N)}, \mathbf{h}_{\pi_1}^{(N)}\right] & t > 1 \\ \left[\bar{\mathbf{h}}^{(N)}, \mathbf{v}^{\mathrm{l}}, \mathbf{v}^{\mathrm{f}}\right] & t = 1. \end{cases} \tag{4}$$

Here $[\cdot, \cdot, \cdot]$ is the horizontal concatenation operator and we write the $(3 \cdot d_{\mathrm{h}})$-dimensional result vector as $\mathbf{h}_{(c)}^{(N)}$ to indicate we interpret it as the embedding of the special context node $(c)$ and use the superscript $(N)$ to align with the node embeddings $\mathbf{h}_i^{(N)}$. We could project the embedding back to $d_{\mathrm{h}}$ dimensions, but we absorb this transformation in the parameter $W^Q$ in equation 5.

Now we compute a new context node embedding $\mathbf{h}_{(c)}^{(N+1)}$ using the ($M$-head) attention mechanism described in Appendix A. The keys and values come from the node embeddings $\mathbf{h}_i^{(N)}$, but we only compute a single query $\mathbf{q}_{(c)}$ (per head) from the context node (we omit the $(N)$ for readability):

$$\mathbf{q}_{(c)} = W^Q \mathbf{h}_{(c)} \quad \mathbf{k}_i = W^K \mathbf{h}_i, \quad \mathbf{v}_i = W^V \mathbf{h}_i. \tag{5}$$

We compute the compatibility of the query with all nodes, and mask (set $u_{(c)j} = -\infty$) nodes which cannot be visited at time $t$. For TSP, this simply means we mask the nodes already visited:

$$u_{(c)j} = \begin{cases} \frac{\mathbf{q}_{(c)}^T \mathbf{k}_j}{\sqrt{d_{\mathrm{k}}}} & \text{if } j \neq \pi_{t'} \quad \forall t' < t \\ -\infty & \text{otherwise.} \end{cases} \tag{6}$$

Here $d_{\mathrm{k}} = \frac{d_{\mathrm{h}}}{M}$ is the query/key dimensionality (see Appendix A). Again, we compute $u_{(c)j}$ and $\mathbf{v}_i$ for $M = 8$ heads and compute the final multi-head attention value for the context node using equations 12–14 from Appendix A, but with $(c)$ instead of $i$. This mechanism is similar to our encoder, but does not use skip-connections, batch normalization or the feed-forward sublayer for maximal efficiency. The result $\mathbf{h}_{(c)}^{(N+1)}$ is similar to the *glimpse* described by Bello et al. (2016).

**Calculation of log-probabilities**  To compute output probabilities $p_{\boldsymbol{\theta}}(\pi_t | s, \boldsymbol{\pi}_{1:t-1})$ in equation 1, we add one final decoder layer with a *single* attention head ($M = 1$ so $d_{\mathrm{k}} = d_{\mathrm{h}}$). For this layer, we *only* compute the compatibilities $u_{(c)j}$ using equation 6, but following Bello et al. (2016) we clip the result (before masking!) within $[-C, C]$ (C = 10) using $\tanh$:

$$u_{(c)j} = \begin{cases} C \cdot \tanh\left(\frac{\mathbf{q}_{(c)}^T \mathbf{k}_j}{\sqrt{d_{\mathrm{k}}}}\right) & \text{if } j \neq \pi_{t'} \quad \forall t' < t \\ -\infty & \text{otherwise.} \end{cases} \tag{7}$$

We interpret these compatibilities as unnormalized log-probabilities (logits) and compute the final output probability vector $\mathbf{p}$ using a softmax (similar to equation 12 in Appendix A):

$$p_i = p_{\boldsymbol{\theta}}(\pi_t = i | s, \boldsymbol{\pi}_{1:t-1}) = \frac{e^{u_{(c)i}}}{\sum_j e^{u_{(c)j}}}. \tag{8}$$

# 4 REINFORCE WITH GREEDY ROLLOUT BASELINE

Section 3 defined our model that given an instance $s$ defines a probability distribution $p_{\boldsymbol{\theta}}(\boldsymbol{\pi}|s)$, from which we can sample to obtain a solution (tour) $\boldsymbol{\pi}|s$. In order to train our model, we define the loss $\mathcal{L}(\boldsymbol{\theta}|s) = \mathbb{E}_{p_{\boldsymbol{\theta}}(\boldsymbol{\pi}|s)}[L(\boldsymbol{\pi})]$: the expectation of the cost $L(\boldsymbol{\pi})$ (tour length for TSP). We optimize $\mathcal{L}$ by gradient descent, using the REINFORCE (Williams, 1992) gradient estimator with baseline $b(s)$:

$$\nabla \mathcal{L}(\boldsymbol{\theta}|s) = \mathbb{E}_{p_{\boldsymbol{\theta}}(\boldsymbol{\pi}|s)}\left[(L(\boldsymbol{\pi}) - b(s)) \nabla \log p_{\boldsymbol{\theta}}(\boldsymbol{\pi}|s)\right]. \tag{9}$$

A good baseline $b(s)$ reduces gradient variance and therefore increases speed of learning. A simple example is an exponential moving average $b(s) = M$ with *decay* $\beta$. Here $M = L(\boldsymbol{\pi})$ in the first iteration and gets updated as $M \leftarrow \beta M + (1-\beta)L(\boldsymbol{\pi})$ in subsequent iterations. A popular alternative is the use of a learned value function (critic) $\hat{v}(s, \boldsymbol{w})$, where the parameters $\boldsymbol{w}$ are learned from the observations $(s, L(\boldsymbol{\pi}))$. However, getting such *actor-critic* algorithms to work is non-trivial.

We propose to use a rollout baseline in a way that is similar to self-critical training by Rennie et al. (2017), but with periodic updates of the baseline policy. It is defined as follows: $b(s)$ is the cost of a solution from a *deterministic greedy rollout* of the policy defined by the best model so far.

**Motivation** The goal of a baseline is to estimate the difficulty of the instance $s$, such that it can relate to the cost $L(\boldsymbol{\pi})$ to estimate the advantage of the solution $\boldsymbol{\pi}$ selected by the model. We make the following key observation: *The difficulty of an instance can (on average) be estimated by the performance of an algorithm applied to it.* This follows from the assumption that (on average) an algorithm will have a higher cost on instances that are more difficult. Therefore we form a baseline by applying (rolling out) the algorithm defined by our model during training. To eliminate variance we force the result to be deterministic by selecting greedily the action with maximum probability.

**Determining the baseline policy** As the model changes during training, we stabilize the baseline by freezing the greedy rollout policy $p_{\boldsymbol{\theta}^{\text{BL}}}$ for a fixed number of steps (every epoch), similar to freezing of the target Q-network in DQN (Mnih et al., 2015). A stronger algorithm defines a stronger baseline, so we compare (with greedy decoding) the current training policy with the baseline policy at the end of every epoch, and replace the parameters $\boldsymbol{\theta}^{\text{BL}}$ of the baseline policy only if the improvement is significant according to a paired t-test ($\alpha = 5\%$), on 10000 separate (evaluation) instances. If the baseline policy is updated, we sample new evaluation instances to prevent overfitting.

**Analysis** With the greedy rollout as baseline $b(s)$, the function $L(\boldsymbol{\pi}) - b(s)$ is negative if the sampled solution $\boldsymbol{\pi}$ is better than the greedy rollout, causing actions to be reinforced, and vice versa. This way the model is trained to improve over its (greedy) self. We see similarities with self-play improvement (Silver et al., 2017): sampling replaces tree search for exploration and the model is rewarded if it yields improvement ('wins') compared to the best model. Similar to AlphaGo, the evaluation at the end of each epoch ensures that we are always challenged by the best model.

**Algorithm** We use Adam (Kingma & Ba, 2015) as optimizer resulting in Algorithm 1.

**Efficiency** Each rollout constitutes an additional forward pass, increasing computation by $50\%$. However, as the baseline policy is fixed for an epoch, we can sample the data and compute baselines per epoch using larger batch sizes, allowed by the reduced memory requirement as the computations can run in pure inference mode. Empirically we find that it adds only $25\%$ (see Appendix B.5), taking up $20\%$ of total time. If desired, the baseline rollout can be computed in parallel such that there is no increase in time per iteration, as an easy way to benefit from an additional GPU.

---

**Algorithm 1** REINFORCE with Rollout Baseline

1: **Input:** number of epochs $E$, steps per epoch $T$, batch size $B$, significance $\alpha$
2: Init $\boldsymbol{\theta}$, $\boldsymbol{\theta}^{\text{BL}} \leftarrow \boldsymbol{\theta}$
3: **for** epoch $= 1, \ldots, E$ **do**
4:      **for** step $= 1, \ldots, T$ **do**
5:          $s_i \leftarrow \text{RandomInstance}()\ \forall i \in \{1, \ldots, B\}$
6:          $\boldsymbol{\pi}_i \leftarrow \text{SampleRollout}(s_i, p_{\boldsymbol{\theta}})\ \forall i \in \{1, \ldots, B\}$
7:          $\boldsymbol{\pi}_i^{\text{BL}} \leftarrow \text{GreedyRollout}(s_i, p_{\boldsymbol{\theta}^{\text{BL}}})\ \forall i \in \{1, \ldots, B\}$
8:          $\nabla \mathcal{L} \leftarrow \sum_{i=1}^{B}\left(L(\boldsymbol{\pi}_i) - L(\boldsymbol{\pi}_i^{\text{BL}})\right) \nabla_{\boldsymbol{\theta}} \log p_{\boldsymbol{\theta}}(\boldsymbol{\pi}_i)$
9:          $\boldsymbol{\theta} \leftarrow \text{Adam}(\boldsymbol{\theta}, \nabla \mathcal{L})$
10:      **end for**
11:      **if** OneSidedPairedTTest$(p_{\boldsymbol{\theta}}, p_{\boldsymbol{\theta}^{\text{BL}}}) < \alpha$ **then**
12:          $\boldsymbol{\theta}^{\text{BL}} \leftarrow \boldsymbol{\theta}$
13:      **end if**
14: **end for**

## 5 Experiments

We focus on routing problems: we consider the TSP, two variants of the VRP, the Orienteering Problem and the (Stochastic) Prize Collecting TSP. These provide a range of different challenges, constraints and objectives and are *traditionally solved by different algorithms*. For the Attention Model (AM), we adjust the input, mask, decoder context and objective function for each problem (see Appendix for details and data generation) and train on problem instances of $n = 20$, 50 and 100 nodes. For all problems, we use *the same hyperparameters*: those we found to work well on TSP.

**Hyperparameters**  We initialize parameters Uniform$(-1/\sqrt{d}, 1/\sqrt{d})$, with $d$ the input dimension. Every epoch we process 2500 batches of 512 instances (except for VRP with $n = 100$, where we use $2500 \times 256$ for memory constraints). For TSP, an epoch takes 5:30 minutes for $n = 20$, 16:20 for $n = 50$ (single GPU 1080Ti) and 27:30 for $n = 100$ (on 2 1080Ti's). We train for 100 epochs using training data generated on the fly. We found training to be stable and results to be robust against different seeds, where only in one case (PCTSP with $n = 20$) we had to restart training with a different seed because the run diverged. We use $N = 3$ layers in the encoder, which we found is a good trade-off between quality of the results and computational complexity. We use a constant learning rate $\eta = 10^{-4}$. Training with a higher learning rate $\eta = 10^{-3}$ is possible and speeds up initial learning, but requires decay (0.96 per epoch) to converge and may be a bit more unstable. See Appendix B.5. With the rollout baseline, we use an exponential baseline ($\beta = 0.8$) during the first epoch, to stabilize initial learning, although in many cases learning also succeeds without this 'warmup'. Our code in PyTorch (Paszke et al., 2017) is publicly available.[2]

**Decoding strategy and baselines**  For each problem, we report performance on 10000 test instances. At test time we use *greedy* decoding, where we select the best action (according to the model) at each step, or *sampling*, where we sample 1280 solutions (in $< 1$s on a single GPU) and report the best. More sampling improves solution quality at increased computation. In Table 1 we compare greedy decoding against baselines that also construct a single solution, and compare sampling against baselines that also consider multiple solutions, either via sampling or (local) search. For each problem, we also report the 'best possible solution': either optimal via Gurobi (2018) (intractable for $n > 20$ except for TSP) or a problem specific state-of-the-art algorithm.

**Run times**  Run times are important but hard to compare: they can vary by two orders of magnitude as a result of implementation (Python vs C++) and hardware (GPU vs CPU). We take a practical view and report the time it takes to solve the test set of 10000 instances, either on a single GPU (1080Ti) or 32 instances in parallel on a 32 virtual CPU system ($2 \times$ Xeon E5-2630). This is conservative: our model is parallelizable while most of the baselines are single thread CPU implementations which cannot parallelize when running individually. Also we note that after training our run time can likely be reduced by model compression (Hinton et al., 2015). In Table 1 we do not report running times for the results which were reported by others as they are not directly comparable but we note that in general our model and implementation is fast: for instance Bello et al. (2016) report 10.3s for sampling 1280 TSP solutions (K80 GPU) which we do in less than one second (on a 1080Ti). For most algorithms it is possible to trade off runtime for performance. As reporting full trade-off curves is impractical we tried to pick reasonable spots, reporting the fastest if results were similar or reporting results with different time limits (for example we use Gurobi with time limits as heuristic).

### 5.1 Problems

**Travelling Salesman Problem (TSP)**  For the TSP, we report optimal results by Gurobi, as well as by Concorde (Applegate et al., 2006) (faster than Gurobi as it is specialized for TSP) and LKH3 (Helsgaun, 2017), a state-of-the-art heuristic solver that empirically also finds optimal solutions in time comparable to Gurobi. We compare against Nearest, Random and Farthest Insertion, as well as Nearest Neighbor, which is the only non-learned baseline algorithm that also constructs a tour directly in order (i.e. is structurally similar to our model). For details, see Appendix B.3. Additionally we compare against the learned heuristics in Section 2, most importantly Bello et al. (2016), as well as OR Tools reported by Bello et al. (2016) and Christofides + 2OPT local search reported by Vinyals

---

[2] https://github.com/wouterkool/attention-learn-to-route

Table 1: Attention Model (AM) vs baselines. The gap % is w.r.t. the best value across all methods.

| | Method | Obj. | $n = 20$ Gap | Time | Obj. | $n = 50$ Gap | Time | Obj. | $n = 100$ Gap | Time |
|---|---|---|---|---|---|---|---|---|---|---|
| TSP | Concorde | 3.84 | 0.00% | (1m) | 5.70 | 0.00% | (2m) | 7.76 | 0.00% | (3m) |
| | LKH3 | 3.84 | 0.00% | (18s) | 5.70 | 0.00% | (5m) | 7.76 | 0.00% | (21m) |
| | Gurobi | 3.84 | 0.00% | (7s) | 5.70 | 0.00% | (2m) | 7.76 | 0.00% | (17m) |
| | Gurobi (1s) | 3.84 | 0.00% | (8s) | 5.70 | 0.00% | (2m) | - | | |
| | Nearest Insertion | 4.33 | 12.91% | (1s) | 6.78 | 19.03% | (2s) | 9.46 | 21.82% | (6s) |
| | Random Insertion | 4.00 | 4.36% | (0s) | 6.13 | 7.65% | (1s) | 8.52 | 9.69% | (3s) |
| | Farthest Insertion | 3.93 | 2.36% | (1s) | 6.01 | 5.53% | (2s) | 8.35 | 7.59% | (7s) |
| | Nearest Neighbor | 4.50 | 17.23% | (0s) | 7.00 | 22.94% | (0s) | 9.68 | 24.73% | (0s) |
| | Vinyals et al. (gr.) | 3.88 | 1.15% | | 7.66 | 34.48% | | - | | |
| | Bello et al. (gr.) | 3.89 | 1.42% | | 5.95 | 4.46% | | 8.30 | 6.90% | |
| | Dai et al. | 3.89 | 1.42% | | 5.99 | 5.16% | | 8.31 | 7.03% | |
| | Nowak et al. | 3.93 | 2.46% | | - | | | - | | |
| | EAN (greedy) | 3.86 | 0.66% | (2m) | 5.92 | 3.98% | (5m) | 8.42 | 8.41% | (8m) |
| | **AM (greedy)** | **3.85** | **0.34%** | (0s) | **5.80** | **1.76%** | (2s) | **8.12** | **4.53%** | (6s) |
| | OR Tools | 3.85 | 0.37% | | 5.80 | 1.83% | | 7.99 | 2.90% | |
| | Chr.f. + 2OPT | 3.85 | 0.37% | | 5.79 | 1.65% | | - | | |
| | Bello et al. (s.) | - | | | 5.75 | 0.95% | | 8.00 | 3.03% | |
| | EAN (gr. + 2OPT) | 3.85 | 0.42% | (4m) | 5.85 | 2.77% | (26m) | 8.17 | 5.21% | (3h) |
| | EAN (sampling) | 3.84 | 0.11% | (5m) | 5.77 | 1.28% | (17m) | 8.75 | 12.70% | (56m) |
| | EAN (s. + 2OPT) | 3.84 | 0.09% | (6m) | 5.75 | 1.00% | (32m) | 8.12 | 4.64% | (5h) |
| | **AM (sampling)** | **3.84** | **0.08%** | (5m) | **5.73** | **0.52%** | (24m) | **7.94** | **2.26%** | (1h) |
| CVRP | Gurobi | 6.10 | 0.00% | | - | | | - | | |
| | LKH3 | 6.14 | 0.58% | (2h) | 10.38 | 0.00% | (7h) | 15.65 | 0.00% | (13h) |
| | RL (greedy) | 6.59 | 8.03% | | 11.39 | 9.78% | | 17.23 | 10.12% | |
| | **AM (greedy)** | **6.40** | **4.97%** | (1s) | **10.98** | **5.86%** | (3s) | **16.80** | **7.34%** | (8s) |
| | RL (beam 10) | 6.40 | 4.92% | | 11.15 | 7.46% | | 16.96 | 8.39% | |
| | Random CW | 6.81 | 11.64% | | 12.25 | 18.07% | | 18.96 | 21.18% | |
| | Random Sweep | 7.08 | 16.07% | | 12.96 | 24.91% | | 20.33 | 29.93% | |
| | OR Tools | 6.43 | 5.41% | | 11.31 | 9.01% | | 17.16 | 9.67% | |
| | **AM (sampling)** | **6.25** | **2.49%** | (6m) | **10.62** | **2.40%** | (28m) | **16.23** | **3.72%** | (2h) |
| SDVRP | RL (greedy) | 6.51 | 4.19% | | 11.32 | 6.88% | | 17.12 | 5.23% | |
| | **AM (greedy)** | **6.39** | **2.34%** | (1s) | **10.92** | **3.08%** | (4s) | **16.83** | **3.42%** | (11s) |
| | RL (beam 10) | 6.34 | 1.47% | | 11.08 | 4.61% | | 16.86 | 3.63% | |
| | **AM (sampling)** | **6.25** | **0.00%** | (9m) | **10.59** | **0.00%** | (42m) | **16.27** | **0.00%** | (3h) |
| OP (distance) | Gurobi | 5.39 | 0.00% | (16m) | - | | | - | | |
| | Gurobi (1s) | 4.62 | 14.22% | (4m) | 1.29 | 92.03% | (6m) | 0.58 | 98.25% | (7m) |
| | Gurobi (10s) | 5.37 | 0.33% | (12m) | 10.96 | 32.20% | (51m) | 1.34 | 95.97% | (53m) |
| | Gurobi (30s) | 5.38 | 0.05% | (14m) | 13.57 | 16.09% | (2h) | 3.23 | 90.28% | (3h) |
| | Compass | 5.37 | 0.36% | (2m) | 16.17 | 0.00% | (5m) | 33.19 | 0.00% | (15m) |
| | Tsili (greedy) | 4.08 | 24.25% | (4s) | 12.46 | 22.94% | (4s) | 25.69 | 22.59% | (5s) |
| | **AM (greedy)** | **5.19** | **3.64%** | (0s) | **15.64** | **3.23%** | (1s) | **31.62** | **4.75%** | (5s) |
| | GA (Python) | 5.12 | 4.88% | (10m) | 10.90 | 32.59% | (1h) | 14.91 | 55.08% | (5h) |
| | OR Tools (10s) | 4.09 | 24.05% | (52m) | - | | | - | | |
| | Tsili (sampling) | 5.30 | 1.62% | (28s) | 15.50 | 4.14% | (2m) | 30.52 | 8.05% | (6m) |
| | **AM (sampling)** | **5.30** | **1.56%** | (4m) | **16.07** | **0.60%** | (16m) | **32.68** | **1.55%** | (53m) |
| PCTSP | Gurobi | 3.13 | 0.00% | (2m) | - | | | - | | |
| | Gurobi (1s) | 3.14 | 0.07% | (1m) | - | | | - | | |
| | Gurobi (10s) | 3.13 | 0.00% | (2m) | 4.54 | 1.36% | (32m) | - | | |
| | Gurobi (30s) | 3.13 | 0.00% | (2m) | 4.48 | 0.03% | (54m) | - | | |
| | **AM (greedy)** | **3.18** | **1.62%** | (0s) | **4.60** | **2.66%** | (2s) | **6.25** | **4.46%** | (5s) |
| | ILS (C++) | 3.16 | 0.77% | (16m) | 4.50 | 0.36% | (2h) | **5.98** | **0.00%** | (12h) |
| | OR Tools (10s) | 3.14 | 0.05% | (52m) | 4.51 | 0.70% | (52m) | 6.35 | 6.21% | (52m) |
| | OR Tools (60s) | **3.13** | **0.01%** | (5h) | **4.48** | **0.00%** | (5h) | 6.07 | 1.56% | (5h) |
| | ILS (Python 10x) | 5.21 | 66.19% | (4m) | 12.51 | 179.05% | (3m) | 23.98 | 300.95% | (3m) |
| | AM (sampling) | 3.15 | 0.45% | (5m) | 4.52 | 0.74% | (19m) | 6.08 | 1.67% | (1h) |
| SPCTSP | REOPT (all) | 3.34 | 2.38% | (17m) | 4.68 | 1.04% | (2h) | 6.22 | 1.10% | (12h) |
| | REOPT (half) | 3.31 | 1.38% | (25m) | **4.64** | **0.00%** | (3h) | **6.16** | **0.00%** | (16h) |
| | REOPT (first) | 3.31 | 1.60% | (1h) | 4.66 | 0.44% | (22h) | - | | |
| | **AM (greedy)** | **3.26** | **0.00%** | (0s) | 4.65 | 0.33% | (2s) | 6.32 | 2.69% | (5s) |

et al. (2015). Results for Dai et al. (2017) are (optimistically) computed from the *optimality gaps* they report on 15-20, 40-50 and 50-100 node graphs, respectively. Using a *single* greedy construction we outperform traditional baselines and we are able to achieve significantly closer to optimal results than previous learned heuristics (from around 1.5% to 0.3% above optimal for $n = 20$). Naturally, the difference with Bello et al. (2016) gets diluted when sampling many solutions (as with many samples even a random policy performs well), but we still obtain significantly better results, *without* tuning the softmax temperature. For completeness, we also report results from running the Encode-Attend-Navigate (EAN) code[3] which is concurrent work by Deudon et al. (2018) (for details see Appendix B.4). Our model outperforms EAN, even if EAN is improved with 2OPT local search. Appendix B.5 presents the results visually, including generalization results for different $n$.

**Vehicle Routing Problem (VRP)**    In the Capacitated VRP (CVRP) (Toth & Vigo, 2014), each node has a demand and multiple routes should be constructed (starting and ending at the depot), such that the total demand of the nodes in each route does not exceed the vehicle capacity. We also consider the Split Delivery VRP (SDVRP), which allows to split customer demands over multiple routes. We implement the datasets described by Nazari et al. (2018) and compare against their Reinforcement Learning (RL) framework and the strongest baselines they report. Comparing greedy decoding, we obtain significantly better results. We cannot directly compare our sampling (1280 samples) to their beam search with size 10 (they do not report sampling or larger beam sizes), but note that our greedy method also outperforms their beam search in most (larger) cases, getting (in <1 second/instance) much closer to LKH3 (Helsgaun, 2017), a state-of-the-art algorithm which found best known solutions to CVRP benchmarks. See Appendix C.4 for greedy example solution plots.

**Orienteering Problem (OP)**    The OP (Golden et al., 1987) is an important problem used to model many real world problems. Each node has an associated *prize*, and the goal is to construct a single tour (starting and ending at the depot) that *maximizes* the sum of prizes of nodes visited while being shorter than a maximum (given) length. We consider the prize distributions proposed in Fischetti et al. (1998): *constant*, *uniform* (in Appendix D.4), and increasing with the *distance* to the depot, which we report here as this is the hardest problem. As 'best possible solution' we report Gurobi (intractable for $n > 20$) and *Compass*, the recent state-of-the-art Genetic Algorithm (GA) by Kobeaga et al. (2018), which is only 2% better than sampling 1280 solutions with our method (objective is maximization). We outperform a Python GA[4] (which seems not to scale), as well the construction phase of the heuristic by Tsiligirides (1984) (comparing greedy or 1280 samples) which is structurally similar to the one learned by our model. OR Tools fails to find feasible solutions in a few percent of the cases for $n > 20$.

**Prize Collecting TSP (PCTSP)**    In the PCTSP (Balas, 1989), each node has not only an associated prize, but also an associated penalty. The goal is to collect at least a *minimum* total prize, while minimizing the total tour length plus the sum of penalties of unvisited nodes. This problem is difficult as an algorithm has to trade off the penalty for not visiting a node with the marginal cost/tour length of visiting (which depends on the other nodes visited), while also satisfying the minimum total prize constraint. We compare against OR Tools with 10 or 60 seconds of local search, as well as open source C++[5] and Python[6] implementations of Iterated Local Search (ILS). Although the Attention Model does not find better solutions than OR Tools with 60s of local search, it finds almost equally good results in significantly less time. The results are also within 2% of the C++ ILS algorithm (but obtained much faster), which was the best open-source algorithm for PCTSP we could find.

**Stochastic PCTSP (SPCTSP)**    The Stochastic variant of the PCTSP (SPCTSP) we consider shows how our model can deal with uncertainty naturally. In the SPCTSP, the *expected* node prize is known upfront, but the real collected prize only becomes known upon visitation. With penalties, this problem is a generalization of the stochastic k-TSP (Ene et al., 2018). Since our model constructs a tour one node at the time, we only need to use the real prizes to compute the remaining prize constraint. By contrast, any algorithm that selects a fixed tour may fail to satisfy the prize constraint so an algorithm *must* be adaptive. As a baseline, we implement an algorithm that plans a tour,

---

[3]https://github.com/MichelDeudon/encode-attend-navigate
[4]https://github.com/mc-ride/orienteering
[5]https://github.com/jordanamecler/PCTSP
[6]https://github.com/rafael2reis/salesman

executes part of it and then re-optimizes using the C++ ILS algorithm. We either execute *all* node visits (so planning additional nodes if the result does not satisfy the prize constraint), *half* of the planned node visits (for $O(\log n)$ replanning iterations) or only the *first* node visit, for maximum adaptivity. We observe that our model outperforms all baselines for $n = 20$. We think that failure to account for uncertainty (by the baselines) in the prize might result in the need to visit one or two additional nodes, which is relatively costly for small instances but relatively cheap for larger $n$. Still, our method is beneficial as it provides competitive solutions at a fraction of the computational cost, which is important in online settings.

## 5.2 ATTENTION MODEL VS. POINTER NETWORK AND DIFFERENT BASELINES

Figure 3 compares the performance of the TSP20 Attention Model (AM) and our implementation of the Pointer Network (PN) during training. We use a validation set of size 10000 with greedy decoding, and compare to using an exponential ($\beta = 0.8$) and a critic (see Appendix B.1) baseline. We used two random seeds and a decaying learning rate of $\eta = 10^{-3} \times 0.96^{\text{epoch}}$. This performs best for the PN, while for the AM results are similar to using $\eta = 10^{-4}$ (see Appendix B.5). This clearly illustrates how the improvement we obtain is the result of both the AM and the rollout baseline: the AM outperforms the PN using any baseline and the rollout baseline improves the quality and convergence speed for both AM and PN. For the PN with critic

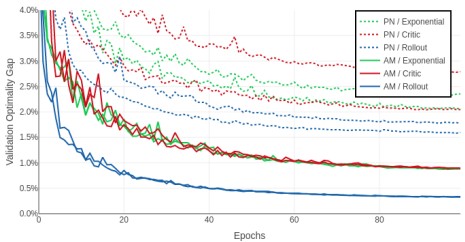

Figure 3: Held-out validation set optimality gap as a function of the number of epochs for the Attention Model (AM) and Pointer Network (PN) with different baselines (two different seeds).

baseline, we are unable to reproduce the $1.5\%$ reported by Bello et al. (2016) (also when using an LSTM based critic), but our reproduction is closer than others have reported (Dai et al., 2017; Nazari et al., 2018). In Table 1 we compare against the original results. Compared to the rollout baseline, the exponential baseline is around 20% faster per epoch, whereas the critic baseline is around 13% slower (see Appendix B.5), so the picture does not change significantly if time is used as x-axis.

## 6 DISCUSSION

In this work we have introduced a model and training method which both contribute to significantly improved results on learned heuristics for TSP and additionally learned strong (single construction) heuristics for multiple routing problems, which are traditionally solved by problem-specific approaches. We believe that our method is a powerful starting point for learning heuristics for other combinatorial optimization problems defined on graphs, if their solutions can be described as sequential decisions. In practice, operational constraints often lead to many variants of problems for which no good (human-designed) heuristics are available such that the ability to learn heuristics could be of great practical value.

Compared to previous works, by using attention instead of recurrence (LSTMs) we introduce invariance to the input order of the nodes, increasing learning efficiency. Also this enables parallelization, for increased computational efficiency. The multi-head attention mechanism can be seen as a message passing algorithm that allows nodes to communicate relevant information over different channels, such that the node embeddings from the encoder can learn to include valuable information about the node *in the context of the graph*. This information is important in our setting where decisions relate directly to the nodes in a graph. Being a graph based method, our model has increased scaling potential (compared to LSTMs) as it can be applied on a sparse graph and operate locally.

Scaling to larger problem instances is an important direction for future research, where we think we have made an important first step by using a graph based method, which can be sparsified for improved computational efficiency. Another challenge is that many problems of practical importance have feasibility constraints that cannot be satisfied by a simple masking procedure, and we think it is promising to investigate if these problems can be addressed by a combination of heuristic learning and backtracking. This would unleash the potential of our method, already highly competitive to the popular Google OR Tools project, to an even larger class of difficult practical problems.

ACKNOWLEDGEMENTS

This research was funded by ORTEC Optimization Technology. We thank Thomas Kipf for helpful discussions and anonymous reviewers for comments that helped improve the paper. We thank DAS5 (Bal et al., 2016) for computational resources and we thank SURFsara (www.surfsara.nl) for the support in using the Lisa Compute Cluster.

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

## A    ATTENTION MODEL DETAILS

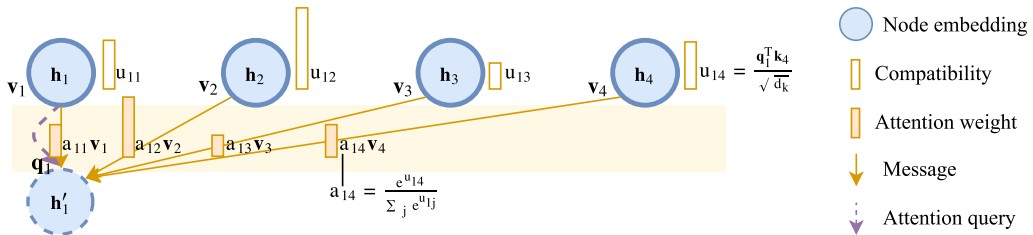

Figure 4: Illustration of weighted message passing using a dot-attention mechanism. Only computation of messages received by node 1 are shown for clarity. Best viewed in color.

**Attention mechanism**    We interpret the attention mechanism by Vaswani et al. (2017) as a weighted message passing algorithm between nodes in a graph. The weight of the message *value* that a node receives from a neighbor depends on the *compatibility* of its *query* with the *key* of the neighbor, as illustrated in Figure 4. Formally, we define dimensions $d_k$ and $d_v$ and compute the key $\mathbf{k}_i \in \mathbb{R}^{d_k}$, value $\mathbf{v}_i \in \mathbb{R}^{d_v}$ and query $\mathbf{q}_i \in \mathbb{R}^{d_k}$ for each node by projecting the embedding $\mathbf{h}_i$:

$$\mathbf{q}_i = W^Q \mathbf{h}_i, \quad \mathbf{k}_i = W^K \mathbf{h}_i, \quad \mathbf{v}_i = W^V \mathbf{h}_i. \tag{10}$$

Here parameters $W^Q$ and $W^K$ are $(d_k \times d_h)$ matrices and $W^V$ has size $(d_v \times d_h)$. From the queries and keys, we compute the compatibility $u_{ij} \in \mathbb{R}$ of the query $\mathbf{q}_i$ of node $i$ with the key $\mathbf{k}_j$ of node $j$ as the (scaled, see Vaswani et al. (2017)) dot-product:

$$u_{ij} = \begin{cases} \frac{\mathbf{q}_i^T \mathbf{k}_j}{\sqrt{d_k}} & \text{if } i \text{ adjacent to } j \\ -\infty & \text{otherwise.} \end{cases} \tag{11}$$

In a general graph, defining the compatibility of non-adjacent nodes as $-\infty$ prevents message passing between these nodes. From the compatibilities $u_{ij}$, we compute the *attention weights* $a_{ij} \in [0, 1]$ using a softmax:

$$a_{ij} = \frac{e^{u_{ij}}}{\sum_{j'} e^{u_{ij'}}}. \tag{12}$$

Finally, the vector $\mathbf{h}'_i$ that is received by node $i$ is the convex combination of messages $\mathbf{v}_j$:

$$\mathbf{h}'_i = \sum_j a_{ij} \mathbf{v}_j. \tag{13}$$

**Multi-head attention**    As was noted by Vaswani et al. (2017) and Velickovic et al. (2018), it is beneficial to have multiple attention heads. This allows nodes to receive different types of messages from different neighbors. Especially, we compute the value in equation 13 $M = 8$ times with different parameters, using $d_k = d_v = \frac{d_h}{M} = 16$. We denote the result vectors by $\mathbf{h}'_{im}$ for $m \in 1, \ldots, M$. These are projected back to a single $d_h$-dimensional vector using $(d_h \times d_v)$ parameter matrices $W^O_m$. The final multi-head attention value for node $i$ is a function of $\mathbf{h}_1, \ldots, \mathbf{h}_n$ through $\mathbf{h}'_{im}$:

$$\text{MHA}_i(\mathbf{h}_1, \ldots, \mathbf{h}_n) = \sum_{m=1}^{M} W^O_m \mathbf{h}'_{im}. \tag{14}$$

**Feed-forward sublayer**    The feed-forward sublayer computes node-wise projections using a hidden (sub)sublayer with dimension $d_{ff} = 512$ and a ReLu activation:

$$\text{FF}(\hat{\mathbf{h}}_i) = W^{\text{ff},1} \cdot \text{ReLu}(W^{\text{ff},0} \hat{\mathbf{h}}_i + \boldsymbol{b}^{\text{ff},0}) + \boldsymbol{b}^{\text{ff},1}. \tag{15}$$

**Batch normalization**    We use batch normalization with learnable $d_h$-dimensional affine parameters $\boldsymbol{w}^{\text{bn}}$ and $\boldsymbol{b}^{\text{bn}}$:

$$\text{BN}(\mathbf{h}_i) = \boldsymbol{w}^{\text{bn}} \odot \overline{\text{BN}}(\mathbf{h}_i) + \boldsymbol{b}^{\text{bn}}. \tag{16}$$

Here $\odot$ denotes the element-wise product and $\overline{\text{BN}}$ refers to batch normalization without affine transformation.

## B  TRAVELLING SALESMAN PROBLEM

### B.1  CRITIC ARCHITECTURE

The critic network architecture uses 3 attention layers similar to our encoder, after which the node embeddings are averaged and processed by an MLP with one hidden layer with 128 neurons and ReLu activation and a single output. We used the same learning rate as for the AM/PN in all experiments.

### B.2  INSTANCE GENERATION

For all TSP instances, the $n$ node locations are sampled uniformly at random in the unit square. This distribution is chosen to be neither easy nor artificially hard and to be able to compare to other learned heuristics.

### B.3  DETAILS OF BASELINES

This section describes details of the heuristics implemented for the TSP. All of the heuristics construct a single tour in a single pass, by extending a partial solution one node at the time.

**Nearest neighbor**   The nearest neighbor heuristic represents the partial solution as a *path* with a *start* and *end* node. The initial path is formed by a single node, selected randomly, which becomes the start node but also the end node of the initial path. In each iteration, the next node is selected as the node nearest to the end node of the partial path. This node is added to the path and becomes the new end node. Finally, after all nodes are added this way, the end node is connected with the start node to form a tour. In our implementation, for deterministic results we always start with the first node in the input, which can be considered random as the instances are generated randomly.

**Farthest/nearest/random insertion**   The insertion heuristics represent a partial solution as a *tour*, and extends it by *inserting* nodes one node at the time. In our implementation, we always insert the node using the *cheapest* insertion cost. This means that when node $i$ is inserted, the place of insertion (between adjacent nodes $j$ and $k$ in the tour) is selected such that it minimizes the *insertion costs* $d_{ji} + d_{ik} - d_{jk}$, where $d_{ji}$, $d_{ik}$ and $d_{jk}$ represent the distances from node $j$ to $i$, $i$ to $k$ and $j$ to $k$, respectively.

The different variants of the insertion heuristic vary in the way in which the node which is inserted is selected. Let $S$ be the set of nodes in the partial tour. *Nearest* insertion inserts the node $i$ that is nearest to (any node in) the tour:

$$i^* = \arg\min_{i \notin S} \min_{j \in S} d_{ij}. \tag{17}$$

*Farthest* insertion inserts the node $i$ such that the distance to the tour (i.e. the distance from $i$ to the nearest node $j$ in the tour) is maximized:

$$i^* = \arg\max_{i \notin S} \min_{j \in S} d_{ij}. \tag{18}$$

*Random* insertion inserts a random node. Similar to nearest neighbor, we consider the input order random so we simply insert the nodes in this order.

### B.4  COMPARISON TO CONCURRENT WORK

Independently of our work, Deudon et al. (2018) also developed a model for TSP based on the Transformer (Vaswani et al., 2017). There are important differences to this paper:

- As 'context' for the decoder, Deudon et al. (2018) use the embeddings of the last $K = 3$ visited nodes. We use only the last (e.g. $K = 1$) node but add the *first* visited node (as well as the graph embedding), since the first node is important (it is the destination) while the order of the other nodes is irrelevant as we explain in Section 3.
- Deudon et al. (2018) use a critic as baseline (which also uses the Transformer architecture). We also experiment with using a critic (based on the Transformer architecture), but found that using a rollout baseline is much more effective (see Section 5).

- Deudon et al. (2018) report results with sampling 128 solutions, with and without 2OPT local search. We report results without 2OPT, using either a single greedy solution or sampling 1280 solutions and additionally show how this directly improves performance compared to Bello et al. (2016).

- By adding 2OPT on top of the best sampled solution, Deudon et al. (2018) show that the model does not produce a local optimum and results can improve by using a 'hybrid' approach of a learned algorithm with local search. This is a nice example of combining learned and traditional heuristics, but it is not compared against using the Pointer Network (Bello et al., 2016) with 2OPT.

- The model of Deudon et al. (2018) uses a higher dimensionality internally in the decoder (for details see their paper). Training is done with 20000 steps with a batch size of 256.

- Deudon et al. (2018) apply Principal Component Analysis (PCA) on the input coordinates to eliminate rotation symmetry whereas we directly input node coordinates.

- Additionally to TSP, we also consider two variants of VRP, the OP with different prize distributions and the (stochastic) PCTSP.

We want to emphasize that this is independent work, but for completeness we include a full emperical comparison of performance. Since the results presented in the paper by Deudon et al. (2018) are not directly comparable, we ran their code[7] and report results under the same circumstances: using greedy decoding and sampling 1280 solutions on our test dataset (which has exactly the same generative procedure, e.g. uniform in the unit square). Additionally, we include results of their model with 2OPT, showing that (even without 2OPT) final performance of our model is better. We use the hyperparameters in their code, but increase the batch size to 512 and number of training steps to $100 \times 2500 = 250000$ for a fair comparison (this increased the performance of their model). As training with $n = 100$ gave out-of-memory errors, we train only on $n = 20$ and $n = 50$ and (following Deudon et al. (2018)) report results for $n = 100$ using the model trained for $n = 50$. The training time as well as test run times are comparable.

## B.5 EXTENDED RESULTS

**Hyperparameters** We found in general that using a larger learning rate of $10^{-3}$ works better with decay but may be unstable in some cases. A smaller learning rate $10^{-4}$ is more stable and does not require decay. This is illustrated in Figure 6, which shows validation results over time using both $10^{-3}$ and $10^{-4}$ with and without decay for TSP20 and TSP50 (2 seeds). As can be seen, without decay the method has not yet fully converged after 100 epochs and results may improve even further with longer training.

Table 2 shows the results in absolute terms as well as the relative *optimality gap* compared to Gurobi, for all runs using seeds 1234 and 1235 with the two different learning rate schedules. We did not run final experiments for $n = 100$ with the larger learning rate as we found training with the smaller learning rate to be more stable. It can be seen that in most cases the end results with different learning rate schedules are similar, except for the larger models ($N = 5$, $N = 8$) where some of the runs diverged using the larger learning rate. Experiments with different number of layers $N$ show that $N = 3$ and $N = 5$ achieve best performance, and we find $N = 3$ is a good trade-off between quality of the results and computational complexity (runtime) of the model.

**Generalization** We test generalization performance on different $n$ than trained for, which we plot in Figure 5 in terms of the relative optimality gap compared to Gurobi. The train sizes are indicated with vertical marker bars. The models generalize when tested on different sizes, although quality degrades as the difference becomes bigger, which can be expected as there is *no free lunch* (Wolpert & Macready, 1997). Since the architectures are the same, these differences mean the models learn to specialize on the problem sizes trained for. We can make a strong overall algorithm by selecting the trained model with highest validation performance for each instance size $n$ (marked in Figure 5 by the red bar). For reference, we also include the baselines, where for the methods that perform search or sampling we do not connect the dots to prevent cluttering and to make the distinction with methods that consider only a single solution clear.

---

[7] `https://github.com/MichelDeudon/encode-attend-navigate`

Table 2: Epoch durations and results and with different seeds and learning rate schedules for TSP.

| | epoch time | $\eta = 10^{-4}$ | | $\eta = 10^{-3} \times 0.96^{\text{epoch}}$ | |
| --- | --- | --- | --- | --- | --- |
| | | seed = 1234 | seed = 1235 | seed = 1234 | seed = 1235 |
| TSP20 | 5:30 | 3.85 (0.34%) | 3.85 (0.29%) | 3.85 (0.33%) | 3.85 (0.32%) |
| TSP50 | 16:20 | 5.80 (1.76%) | 5.79 (1.66%) | 5.81 (2.02%) | 5.81 (2.00%) |
| TSP100 (2GPUs) | 27:30 | 8.12 (4.53%) | 8.10 (4.34%) | - | - |
| N = 0 | 3:10 | 4.24 (10.50%) | 4.26 (10.95%) | 4.25 (10.79%) | 4.24 (10.55%) |
| N = 1 | 3:50 | 3.87 (0.97%) | 3.87 (1.01%) | 3.87 (0.90%) | 3.87 (0.89%) |
| N = 2 | 5:00 | 3.85 (0.40%) | 3.85 (0.44%) | 3.85 (0.38%) | 3.85 (0.39%) |
| N = 3 | 5:30 | 3.85 (0.34%) | 3.85 (0.29%) | 3.85 (0.33%) | 3.85 (0.32%) |
| N = 5 | 7:00 | 3.85 (0.25%) | 3.85 (0.28%) | 3.85 (0.30%) | 10.43 (171.82%) |
| N = 8 | 10:10 | 3.85 (0.28%) | 3.85 (0.33%) | 10.43 (171.82%) | 10.43 (171.82%) |
| AM / Exponential | 4:20 | 3.87 (0.95%) | 3.87 (0.93%) | 3.87 (0.90%) | 3.87 (0.87%) |
| AM / Critic | 6:10 | 3.87 (0.96%) | 3.87 (0.97%) | 3.87 (0.88%) | 3.87 (0.88%) |
| AM / Rollout | 5:30 | 3.85 (0.34%) | 3.85 (0.29%) | 3.85 (0.33%) | 3.85 (0.32%) |
| PN / Exponential | 5:10 | 3.95 (2.94%) | 3.94 (2.80%) | 3.92 (2.09%) | 3.93 (2.37%) |
| PN / Critic | 7:30 | 3.95 (3.00%) | 3.95 (2.93%) | 3.91 (2.01%) | 3.94 (2.84%) |
| PN / Rollout | 6:40 | 3.93 (2.46%) | 3.93 (2.36%) | 3.90 (1.63%) | 3.90 (1.78%) |

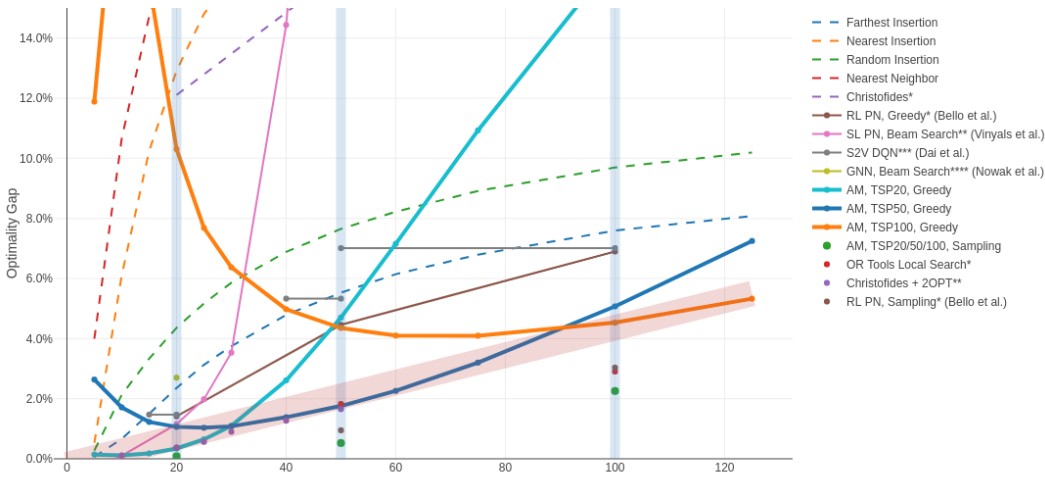

Figure 5: Optimality gap of different methods as a function of problem size $n \in \{5, 10, 15, 20, 25, 30, 40, 50, 60, 75, 100, 125\}$. General baselines are drawn using dashed lines while learned algorithms are drawn with a solid line. Algorithms (general and learned) that perform search or sampling are plotted without connecting lines for clarity. The *, **, *** and **** indicate that values are reported from Bello et al. (2016), Vinyals et al. (2015), Dai et al. (2017) and Nowak et al. (2017) respectively. Best viewed in color.

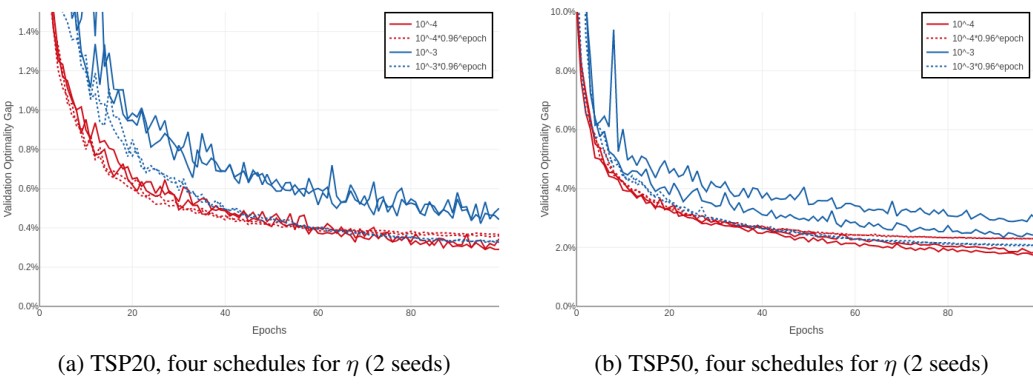

(a) TSP20, four schedules for $\eta$ (2 seeds)   (b) TSP50, four schedules for $\eta$ (2 seeds)

Figure 6: Validation set optimality gap as a function of the number of epochs for different $\eta$.

## C  VEHICLE ROUTING PROBLEM

The Capacitated Vehicle Routing Problem (CVRP) is a generalization of the TSP in which case there is a depot and multiple routes should be created, each starting and ending at the depot. In our graph based formulation, we add a special depot node with index 0 and coordinates $\mathbf{x}_0$. A vehicle (route) has capacity $D > 0$ and each (regular) node $i \in \{1, \dots n\}$ has a demand $0 < \delta_i \leq D$. Each route starts and ends at the depot and the total demand in each route should not exceed the capacity, so $\sum_{i \in R_j} \delta_i \leq D$, where $R_j$ is the set of node indices assigned to route $j$. Without loss of generality, we assume a normalized $\hat{D} = 1$ as we can use normalized demands $\hat{\delta}_i = \frac{\delta_i}{D}$.

The Split Delivery VRP (SDVRP) is a generalization of CVRP in which every node can be visited multiple times, and only a subset of the demand has to be delivered at each visit. Instances for both CVRP and SDVRP are specified in the same way: an instance with size $n$ as a depot location $\mathbf{x}_0$, $n$ node locations $\mathbf{x}_i, i = 1 \dots n$ and (normalized) demands $0 < \hat{\delta}_i \leq 1, i = 1 \dots n$.

### C.1  INSTANCE GENERATION

We follow Nazari et al. (2018) in the generation of instances for $n = 20, 50, 100$, but normalize the demands by the capacities. The depot location as well as $n$ node locations are sampled uniformly at random in the unit square. The demands are defined as $\hat{\delta}_i = \frac{\delta_i}{D^n}$ where $\delta_i$ is discrete and sampled uniformly from $\{1, \dots, 9\}$ and $D^{20} = 30$, $D^{50} = 40$ and $D^{100} = 50$.

### C.2  ATTENTION MODEL FOR THE VRP

**Encoder**  In order to allow our Attention Model to distinguish the depot node from the regular nodes, we use separate parameters $W_0^{\mathrm{x}}$ and $\mathbf{b}_0^{\mathrm{x}}$ to compute the initial embedding $\mathbf{h}_0^{(0)}$ of the depot node. Additionally, we provide the normalized demand $\delta_i$ as input feature (and adjust the size of parameter $W^{\mathrm{x}}$ accordingly):

$$\mathbf{h}_i^{(0)} = \begin{cases} W_0^{\mathrm{x}} \mathbf{x}_i + \mathbf{b}_0^{\mathrm{x}} & i = 0 \\ W^{\mathrm{x}} \left[ \mathbf{x}_i, \hat{\delta}_i \right] + \mathbf{b}^{\mathrm{x}} & i = 1, \dots, n. \end{cases} \tag{19}$$

**Capacity constraints**  To facilitate the capacity constraints, we keep track of the remaining demands $\hat{\delta}_{i,t}$ for the nodes $i \in \{1, \dots n\}$ and remaining vehicle capacity $\hat{D}_t$ at time $t$. At $t = 1$, these are initialized as $\hat{\delta}_{i,t} = \hat{\delta}_i$ and $\hat{D}_t = 1$, after which they are updated as follows (recall that $\pi_t$ is the index of the node selected at decoding step $t$):

$$\hat{\delta}_{i,t+1} = \begin{cases} \max(0, \hat{\delta}_{i,t} - \hat{D}_t) & \pi_t = i \\ \hat{\delta}_{i,t} & \pi_t \neq i \end{cases} \tag{20}$$

$$\hat{D}_{t+1} = \begin{cases} \max(\hat{D}_t - \hat{\delta}_{\pi_t,t}, 0) & \pi_t \neq 0 \\ 1 & \pi_t = 0. \end{cases} \tag{21}$$

If we do not allow split deliveries, $\hat{\delta}_{i,t}$ will be either 0 or $\hat{\delta}_i$ for all $t$.

**Decoder context**  The context for the decoder for the VRP at time $t$ is the current/last location $\pi_{t-1}$ and the remaining capacity $\hat{D}_t$. Compared to TSP, we do not need placeholders if $t = 1$ as the route starts at the depot and we do not need to provide information about the first node as the route should end at the depot:

$$\mathbf{h}_{(c)}^{(N)} = \begin{cases} \left[ \bar{\mathbf{h}}^{(N)}, \mathbf{h}_{\pi_{t-1}}^{(N)}, \hat{D}_t \right] & t > 1 \\ \left[ \bar{\mathbf{h}}^{(N)}, \mathbf{h}_0^{(N)}, \hat{D}_t \right] & t = 1. \end{cases} \tag{22}$$

**Masking**  The depot can be visited multiple times, but we do not allow it to be visited at two subsequent timesteps. Therefore, in both layers of the decoder, we change the masking for the depot $j = 0$ and define $u_{(c)0} = -\infty$ if (and only if) $t = 1$ or $\pi_{t-1} = 0$. The masking for the nodes

depends on whether we allow split deliveries. Without split deliveries, we do not allow nodes to be visited if their remaining demand is 0 (if the node was already visited) or exceeds the remaining capacity, so for $j \neq 0$ we define $u_{(c)j} = -\infty$ if (and only if) $\hat{\delta}_{i,t} = 0$ or $\hat{\delta}_{i,t} > \hat{D}_t$. With split deliveries, we only forbid delivery when the remaining demand is 0, so we define $u_{(c)j} = -\infty$ if (and only if) $\hat{\delta}_{i,t} = 0$.

**Split deliveries**   Without split deliveries, the remaining demand $\hat{\delta}_{i,t}$ is either 0 or $\hat{\delta}_i$, corresponding to whether the node has been visited or not, and this information is conveyed to the model via the masking of the nodes already visited. However, when split deliveries are allowed, the remaining demand $\hat{\delta}_{i,t}$ can take any value $0 \leq \hat{\delta}_{i,t} \leq \hat{\delta}_i$. This information cannot be included in the context node as it corresponds to individual nodes. Therefore we include it in the computation of the keys and values in both the attention layer (glimpse) and the output layer of the decoder, such that we compute queries, keys and values using:

$$\mathbf{q}_{(c)} = W^Q \mathbf{h}_{(c)} \quad \mathbf{k}_i = W^K \mathbf{h}_i + W_d^K \hat{\delta}_{i,t}, \quad \mathbf{v}_i = W^V \mathbf{h}_i + W_d^V \hat{\delta}_{i,t}. \tag{23}$$

Here we $W_d^K$ and $W_d^V$ are $(d_k \times 1)$ parameter matrices and we define $\hat{\delta}_{i,t} = 0$ for the depot $i = 0$. Summing the projection of both $\mathbf{h}_i$ and $\hat{\delta}_{i,t}$ is equivalent to projecting the concatenation $[\mathbf{h}_i, \hat{\delta}_{i,t}]$ with a single $((d_h + 1) \times d_k)$ matrix $W^K$. However, using this formulation we only need to compute the first term once (instead for every $t$) and by the weight initialization this puts more importance on $\hat{\delta}_{i,t}$ initially (which is otherwise just 1 of $d_h + 1 = 129$ input values).

**Training**   For the VRP, the length of the output of the model depends on the number of times the depot is visited. In general, the depot is visited multiple times, and in the case of SDVRP also some regular nodes are visited twice. Therefore the length of the solution is larger than $n$, which requires more memory such that we find it necessary to limit the batch size $B$ to 256 for $n = 100$ (on 2 GPUs). To keep training times tractable and the total number of parameter updates equal, we still process 2500 batches per epoch, for a total of 0.64M training instances per epoch.

## C.3   Details of baselines

For LKH3[8] by Helsgaun (2017) we build and run their code with the `SPECIAL` parameter as specified in their CVRP runscript[9]. We perform 1 run with a maximum of 10000 trials, as we found performing 10 runs only marginally improves the quality of the results while taking much more time.

## C.4   Example solutions

Figure 7 shows example solutions for the CVRP with $n = 100$ that were obtained by a single construction using the model with greedy decoding. These visualizations give insight in the heuristic that the model has learned. In general we see that the model constructs the routes from the bottom to the top, starting below the depot. Most routes are densely packed, except for the last route that has to serve some remaining (close to each other) customers. In most cases, the node in the route that is farthest from the depot is somewhere in the middle of the route, such that customers are served on the way to and from the farthest nodes. In some cases, we see that the order of stops within some individual routes is suboptimal, which means that the method will likely benefit from simple further optimizations on top, such as a beam search, a post-processing procedure based on local search (e.g. 2OPT) or solving the individual routes using a TSP solver.

---

[8] `http://akira.ruc.dk/~keld/research/LKH-3/`
[9] `run_CVRP` in `http://akira.ruc.dk/~keld/research/LKH-3/BENCHMARKS/CVRP.tgz`

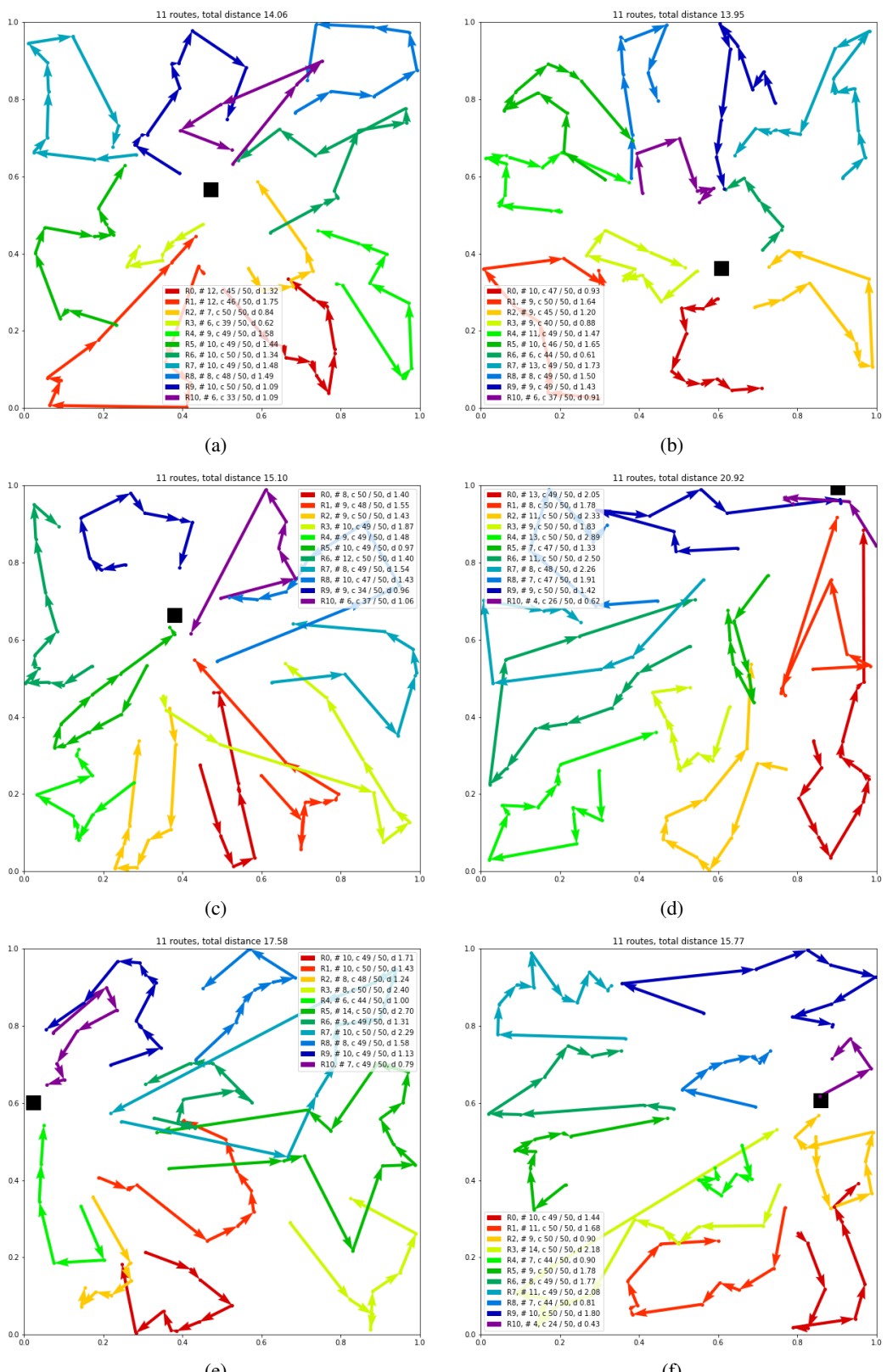

Figure 7: Example greedy solutions for the CVRP ($n = 100$). Edges from and to depot omitted for clarity. Legend order/coloring and arcs indicate the order in which the solution was generated. Legends indicate the number of stops, the used and available capacity and the distance per route.

# D ORIENTEERING PROBLEM

In the Orienteering Problem (OP) each node has a prize $\rho_i$ and the goal is to *maximize* the total prize of nodes visited, while keeping the total length of the route below a maximum length $T$. This problem is different from the TSP and the VRP because visiting each node is optional. Similar to the VRP, we add a special depot node with index 0 and coordinates $\mathbf{x}_0$. If the model selects the depot, we consider the route to be finished. In order to prevent infeasible solutions, we only allow to visit a node if after visiting that node a return to the depot is still possible within the maximum length constraint. Note that it is always suboptimal to visit the depot if additional nodes can be visited, but we do not enforce this knowledge.

## D.1 INSTANCE GENERATION

The depot location as well as $n$ node locations are sampled uniformly at random in the unit square. For the distribution of the prizes, we consider three different variants described by Fischetti et al. (1998), but we normalize the prizes $\rho_i$ such that the normalized prizes $\hat{\rho}_i$ are between 0 and 1.

**Constant**    $\rho_i = \hat{\rho}_i = 1$. Every node has the same prize so the goal becomes to visit as many nodes as possible within the length constraint.

**Uniform**    $\rho_i \sim \text{DiscreteUniform}(1, 100), \hat{\rho}_i = \frac{\rho_i}{100}$. Every node has a prize that is (discretized) uniform.

**Distance**    $\rho_i = 1 + \left\lfloor 99 \cdot \frac{d_{0i}}{\max_{j=1}^n d_{0j}} \right\rfloor, \hat{\rho}_i = \frac{\rho_i}{100}$, where $d_{0i}$ is the distance from the depot to node $i$. Every node has a (discretized) prize that is proportional to the distance to the depot. This is designed to be challenging as the largest prizes are furthest away from the depot (Fischetti et al., 1998).

The maximum length $T^n$ for instances with $n$ nodes (and a depot) is chosen to be (on average) approximately half of the length of the average TSP tour for uniform TSP instances with $n$ nodes[10]. This idea is that this way approximately (a little more than) half of the nodes can be visited, which results in the most difficult problem instances (Vansteenwegen et al., 2011). This is because the number of possible node selections $\binom{n}{k}$ is maximized if $k = \frac{n}{2}$ and additionally determining the actual path is harder with more nodes selected. We set fixed maximum lengths $T^{20} = 2$, $T^{50} = 3$ and $T^{100} = 4$ instead of adjusting the constraint per instance, such that for some instances more or less nodes can be visited. Note that $T^n$ has the same unit as the node coordinates $\mathbf{x}_i$, so we do not normalize them.

## D.2 ATTENTION MODEL FOR THE OP

**Encoder**    Similar to the VRP, we use separate parameters for the depot node embedding. Additionally, we provide the node prize $\hat{\rho}_i$ as input feature:

$$\mathbf{h}_i^{(0)} = \begin{cases} W_0^{\text{x}} \mathbf{x}_i + \mathbf{b}_0^{\text{x}} & i = 0 \\ W^{\text{x}} [\mathbf{x}_i, \hat{\rho}_i] + \mathbf{b}^{\text{x}} & i = 1, \dots, n. \end{cases} \tag{24}$$

**Max length constraint**    In order to satisfy the max length constraint, we keep track of the *remaining* max length $T_t$ at time $t$. Starting at $t = 1$, $T_1 = T$. Then for $t > 0$, $T$ is updated as

$$T_{t+1} = T_t - d_{\pi_{t-1}, \pi_t}. \tag{25}$$

Here $d_{\pi_{t-1}, \pi_t}$ is the distance from node $\pi_{t-1}$ to $\pi_t$ and we conveniently define $\pi_0 = 0$ as we start at the depot.

**Decoder context**    The context for the decoder for the OP at time $t$ is the current/last location $\pi_{t-1}$ and the remaining max length $T_t$. Similar to VRP, we do not need placeholders if $t = 1$ as the route starts at the depot and we do not need to provide information about the first node as the route should

---

[10]The average length of the optimal TSP tour is 3.84, 5.70 and 7.76 for $n = 20, 50, 100$.

end at the depot. We do not need to provide information on the prizes gathered as this is irrelevant for the remaining decisions. The context is defined as:

$$\mathbf{h}_{(c)}^{(N)} = \begin{cases} \left[\bar{\mathbf{h}}^{(N)}, \mathbf{h}_{\pi_{t-1}}^{(N)}, T_t\right] & t > 1 \\ \left[\bar{\mathbf{h}}^{(N)}, \mathbf{h}_0^{(N)}, T_t\right] & t = 1. \end{cases} \tag{26}$$

**Masking** In the OP, the depot node can always be visited so is never masked. Regular nodes are masked (i.e. cannot be visited) if either they are already visited or if they cannot be visited within the remaining length constraint:

$$u_{(c)j} = -\infty \Leftrightarrow \exists t' < t : \pi_{t'} = j \text{ or } d_{\pi_{t-1},j} + d_{j0} > T_t \tag{27}$$

### D.3 Details of baselines

For Compass[11] by Kobeaga et al. (2018), we compile their code and run it with default parameters, only adding `--op --op-ea4op` to indicate that the Genetic Algorithm for the Orienteering Problem should be used. As Compass uses integer coordinates and prizes, we multiply all floats by $10^7$ and round to integers. We run the Python Genetic Algorithm[12] with default parameters.

**Tsiligirides** Tsiligirides (1984) describes a heuristic procedure for solving the OP. It consists of sampling 3000 tours through a randomized construction procedure and applies local search on top. The randomized construction part of the heuristic is structurally exactly the same as the heuristic learned by our model, but with a manually engineered function to define the node probabilities. We implement the construction part of the heuristic and compare it to our model (either greedy or sampling 1280 solutions), without the local search (as this can also be applied on top of our model). The final heuristic used by Tsiligirides (1984) uses a formula with multiple terms to define the probability that a node should be selected, but by tuning the weights the form with only one simple term works best, showing the difficulty of manually defining a good probability distribution. In our terms, the heuristic defines a score $s_i$ for each node at time $t$ as the prize divided by the distance from the current node $\pi_{t-1}$, raised to the 4th power:

$$s_i = \left(\frac{\hat{\rho}_i}{d_{\pi_{t-1},i}}\right)^4. \tag{28}$$

Let $S$ be the set with the $\min(4, n - (t-1))$ unvisited nodes with maximum score $s_i$. Then the node probabilities $p_i$ at time $t$ are defined as

$$p_i = p_{\boldsymbol{\theta}}(\pi_t = i | s, \boldsymbol{\pi}_{1:t-1}) = \begin{cases} \frac{s_i}{\sum_{j \in S} s_j} & \text{if } i \in S \\ 0 & \text{otherwise.} \end{cases} \tag{29}$$

**OR Tools** For the Google OR Tools implementation, we modify the formulation for the CVRP[13]:

- We replace the Manhattan distance by the Euclidian distance.
- We set the number of vehicles to 1.
- For each individual node $i$, we add a *Disjunction constraint* with $\{i\}$ as the set of nodes, and a penalty equal to the prize $\hat{\rho}_i$. This allows OR tools to skip node $i$ at a cost $\hat{\rho}_i$.
- We replace the capacity constraint by a maximum distance. constraint
- We remove the objective to minimize the length.

We multiply all float inputs by $10^7$ and round to integers. Note that OR Tools computes penalties for skipped nodes rather than gains for nodes that are visited. The problem is equivalent, but in order to compare the objective value against our method, we need to add the constant sum of all penalties $\sum_i \hat{\rho}_i$ to the OR Tools objective.

---

[11]https://github.com/bcamath-ds/compass
[12]https://github.com/mc-ride/orienteering
[13]https://github.com/google/or-tools/blob/master/examples/python/cvrp.py

## D.4 Extended results

Table 3 displays the results for the OP with constant and uniform prize distributions. The results are similar to the results for the prize distribution based on the distance to the depot, although by the calculation time for Gurobi it is confirmed that indeed constant and uniform prize distributions are easier.

## E Prize Collecting TSP

In the Prize Collecting TSP (PCTSP) each node has a prize $\rho_i$ and an associated penalty $\beta_i$. The goal is to minimize the total length of the tour plus the sum of penalties for nodes which are not visited, while collecting at least a given minimum total prize. W.l.o.g. we assume the minimum total prize is equal to 1 (as prizes can be normalized). This problem is related to the OP but inverts the goal (minimizing tour length given a minimum total prize to collect instead of maximizing total prize given a maximum tour length) and additionally adds penalties. Again, we add a special depot node with index 0 and coordinates $\mathbf{x}_0$ and if the model selects the depot, the route is finished. In the PCTSP, it can be beneficial to visit additional nodes, even if the minimum total prize constraint is already satisfied, in order to avoid penalties.

Table 3: Additional results for the OP

| Method | 20 | | 50 | | 100 | |
|---|---|---|---|---|---|---|
| **OP (constant)** | | | | | | |
| Gurobi | 10.57 | (4m) | - | | - | |
| Compass | 10.56 | (55s) | 29.58 | (3m) | 59.35 | (8m) |
| Tsili (greedy) | 8.82 | (5s) | 23.89 | (4s) | 47.65 | (5s) |
| AM (greedy) | **10.27** | (0s) | **28.31** | (2s) | **55.81** | (5s) |
| GA (Python) | 9.72 | (10m) | 18.52 | (1h) | 25.68 | (5h) |
| OR Tools (10s) | 8.54 | (52m) | - | | - | |
| Tsili (sampling) | 10.48 | (28s) | 28.26 | (2m) | 54.27 | (6m) |
| AM (sampling) | **10.49** | (4m) | **29.36** | (17m) | **58.33** | (56m) |
| **OP (uniform)** | | | | | | |
| Gurobi | 5.85 | (7m) | - | | - | |
| Compass | 5.84 | (1m) | 16.46 | (5m) | 33.30 | (14m) |
| Tsili (greedy) | 4.85 | (4s) | 12.80 | (4s) | 25.48 | (5s) |
| AM (greedy) | **5.60** | (0s) | **15.62** | (2s) | **31.03** | (5s) |
| GA (Python) | 5.53 | (10m) | 10.81 | (1h) | 14.89 | (5h) |
| OR Tools (10s) | 4.69 | (52m) | - | | - | |
| Tsili (sampling) | 5.70 | (26s) | 15.28 | (2m) | 29.54 | (5m) |
| AM (sampling) | **5.76** | (4m) | **16.25** | (16m) | **32.41** | (51m) |

### E.1 Instance generation

The depot location as well as $n$ node locations are sampled uniformly at random in the unit square. Similar to the OP, we select the distribution for the prizes and penalties with the idea that for difficult instances approximately half of the nodes should be visited. Additionally, neither the prize nor the penalty should dominate the node selection process.

**Prizes** We consider uniformly distributed prizes. If we sample prizes $\rho_i \sim \mathrm{Uniform}(0, 1)$, then $\mathbb{E}(\rho_i) = \frac{1}{2}$, and the expected total prize of any subset of $\frac{n}{2}$ nodes (i.e. half of the nodes) would be $\frac{n}{4}$. Therefore, if $S$ is the set of nodes that is visited, we require that $\sum_{i \in S} \rho_i \geq \frac{n}{4}$, or equivalently $\sum_{i \in S} \hat{\rho}_i \geq 1$ where $\hat{\rho}_i = \rho_i \cdot \frac{4}{n}$ is the normalized prize. Note that it can be the case that $\sum_{i=1}^{n} \hat{\rho}_i < 1$, in which case the prize constraint may be violated but it is only allowed to return to the depot after all nodes have been visited.

**Penalties** If penalties are too small, then node selection is determined almost entirely by the minimum total prize constraint. If penalties are too large, we will always visit all nodes, making the minimum total prize constraint obsolete. We argue that in order for the penalties to be meaningful, they should contribute a term in the objective approximately equal to the total length of the tour. If $L^n$ is the expected TSP tour length with $n$ nodes, we try to achieve this by sampling $\beta_i \sim \mathrm{Uniform}(0, 2 \cdot \frac{L^n}{n})$ such that $\mathbb{E}(\beta_i) = \frac{L^n}{n}$ and the expected total penalty for a subset of $\frac{n}{2}$ nodes is $\frac{L^n}{2}$. Following the numbers we use for the OP, we roughly define $\frac{L^n}{2} \approx K^n = 2, 3, 4$ for $n = 20, 50, 100$[14]. This means that we should sample $\beta_i \sim \mathrm{Uniform}(0, 4 \cdot \frac{K^n}{n})$, but empirically we find that $\hat{\beta}_i \sim \mathrm{Uniform}(0, 3 \cdot \frac{K^n}{n})$ works better, which means that the prizes and penalties are balanced as the minimum total prize constraint is sometimes binding and sometimes not.

---

[14] The average length of the optimal TSP tour is 3.84, 5.70 and 7.76 for $n = 20, 50, 100$.

## E.2 ATTENTION MODEL FOR THE PCTSP

**Encoder** Again, we use separate parameters for the depot node embedding. Additionally, we provide the node prize $\hat{\rho}_i$ and the penalty $\hat{\beta}_i$ as input features:

$$\mathbf{h}_i^{(0)} = \begin{cases} W_0^{\mathrm{x}} \mathbf{x}_i + \mathbf{b}_0^{\mathrm{x}} & i = 0 \\ W^{\mathrm{x}} \left[ \mathbf{x}_i, \hat{\rho}_i, \hat{\beta}_i \right] + \mathbf{b}^{\mathrm{x}} & i = 1, \dots, n. \end{cases} \tag{30}$$

**Minimum prize constraint** In order to satisfy the minimum total prize constraint, we keep track of the *remaining* total prize $P_t$ to collect at time $t$. At $t = 1$, $P_1 = 1$ (as we normalized prizes). Then for $t > 0$, $P$ is updated as

$$P_{t+1} = \max(0, P_t - \hat{\rho}_{\pi_t}). \tag{31}$$

If the constraint is satisfied after visiting $\pi_t$ is visited at time $t$, then $P_{t+1}$ will be 0.

**Decoder context** The context for the decoder for the PCTSP at time $t$ is the current/last location $\pi_{t-1}$ and the remaining prize to collect $P_t$. Again, we do not need placeholders if $t = 1$ as the route starts at the depot and we do not need to provide information about the first node as the route should end at the depot. The information about the prizes collected is implicitly provided to the model in the form of $P_t$ and we do not need to provide any information about the penalties as this is irrelevant for the remaining decisions:

$$\mathbf{h}_{(c)}^{(N)} = \begin{cases} \left[ \bar{\mathbf{h}}^{(N)}, \mathbf{h}_{\pi_{t-1}}^{(N)}, P_t \right] & t > 1 \\ \left[ \bar{\mathbf{h}}^{(N)}, \mathbf{h}_0^{(N)}, P_t \right] & t = 1. \end{cases} \tag{32}$$

**Masking** In the PCTSP, the depot node cannot be visited if the remaining prize to collect $P_t$ is larger than 0 and not yet all nodes have been visited (so $t \leq n$):

$$u_{(c)0} = -\infty \Leftrightarrow P_t > 0 \text{ and } t \leq n. \tag{33}$$

Regular nodes are masked (i.e. cannot be visited) only if they are already visited:

$$u_{(c)j} = -\infty \Leftrightarrow \exists t' < t : \pi_{t'} = j. \tag{34}$$

## E.3 DETAILS OF BASELINES

For the C++ Iterated Local Search (ILS) algorithm[15], we perform 1 run as this takes already 2 minutes per instance (single thread) on average. For the Python ILS algorithm[16] we perform 10 runs as this algorithm is fast. This improved results somewhat for $n = 20$.

**OR Tools** For the Google OR Tools implementation, we modify the formulation for the CVRP[17]:

- We replace the Manhattan distance by the Euclidian distance.
- We set the number of vehicles to 1.
- For each individual node $i$, we add a *Disjunction constraint* with $\{i\}$ as the set of nodes, and a penalty equal to the penalty $\hat{\beta}_i$. This allows OR tools to skip node $i$ at a cost $\hat{\beta}_i$.
- We replace the capacity constraint by a minimum total prize constraint by adding the prizes as a *Dimension*.

We multiply all float inputs by $10^7$ and round to integers. Note that we keep the total length objective from the CVRP and add the Disjunction constraint with penalties to obtain the right objective.

---

[15]https://github.com/jordanamecler/PCTSP
[16]https://github.com/rafael2reis/salesman
[17]https://github.com/google/or-tools/blob/master/examples/python/cvrp.py

# F   STOCHASTIC PCTSP (SPCTSP)

For the SPCTSP, we assume that the real prize collected $\hat{\rho}_i^*$ at each node only becomes known when visiting the node, and $\hat{\rho}_i = \mathbb{E}\left[\hat{\rho}_i^*\right]$ is the expected prize. We assume the real prizes follow a uniform distribution, so $\hat{\rho}_i^* \sim \text{Uniform}(0, 2\hat{\rho}_i)$.

## F.1   ATTENTION MODEL FOR THE SPCTSP

In order to apply the Attention Model to the Stochastic PCTSP, the only change we need is that we use the real $\hat{\rho}_i^*$ to update the remaining prize to collect $P_t$ in equation 31:

$$P_{t+1} = \max(0, P_t - \hat{\rho}_{\pi_t}^*). \tag{35}$$

We could theoretically use the model trained for PCTSP without retraining, but we choose to retrain. This way the model could (for example) learn that *if* it needs to gather a remaining (normalized) prize of $0.1$, it might prefer to visit a node with expected prize $0.2$ over a node with expected prize $0.1$ as the first real prize will be $\geq 0.1$ with probability $75\%$ (uniform prizes) whereas the latter only with $50\%$ and thus has a probability of $50\%$ to not satisfy the constraint.

## F.2   ROLLOUT BASELINE IN THE STOCHASTIC SETTING

Instead of sampling the real prizes online, we already sample them when creating the dataset but keep them hidden to the algorithm. This way, when using a rollout baseline, both the greedy rollout baseline as well as the sample (rollout) from the model use the same real prizes, such that any difference between the two is not a result of stochasticity. This can be seen as a variant of using Common Random Numbers for variance reduction (Glasserman & Yao, 1992).

## F.3   DETAILS OF BASELINES

For the SPCTSP, it is not possible to formulate an exact model that constructs a tour offline (as any tour can be infeasible with nonzero probability) and an algorithm that computes the optimal decision online should take into account an infinite number of scenarios. As a baseline we implement a strategy that:

1. Plans a tour using the expected prizes $\hat{\rho}_i$
2. Executes part of the tour (not returning to the depot), observing the real prizes $\hat{\rho}_i^*$
3. Computes the remaining total prize that needs to be collected
4. Computes a new tour (again using expected prizes $\hat{\rho}_i$), starting from the last node that was visited, through nodes that have not yet been visited and ending at the depot
5. Repeats the steps (2) - (4) above until the minimum total prize has been collected or all nodes have been visited
6. Returns to the depot

Planning of the tours using deterministic prizes means we need to solve a (deterministic) PCTSP, for which we use the ILS C++ algorithm as this was the strongest algorithm for PCTSP (for large $n$). Note that in (4), we have a variant of the PCTSP where we do not have a single depot, but rather separate start and end points, whereas the ILS C++ implementation assumes starting and ending at a single depot. However, as the ILS C++ implementation uses a distance matrix, we can effectively plan with a start and end node by defining the distance from the 'depot' to node $j$ as the distance from the start node (the last visited node) to node $j$, whereas we leave the distance from node $j$ to the depot/end node unchanged (so the distance matrix becomes asymmetrical). Additionally, we remove all nodes (rows/columns in the distance matrix) that have already been visited from the problem.

We consider three variants that differ in the number of nodes that are visited before replanning the tour, for a tradeoff between adaptivity and run time:

1. *All* nodes in the planned tour are visited (except the final return to the depot). We only need to replan and visit additional nodes if the constraint is not satisfied, otherwise we return to the depot.

2. *Half* of the nodes *in the planned tour* are visited, where we visit $k$ nodes if there are $2k+1$ nodes (excluding the return to the depot), so we round down if an odd number of visits is planned. This way, we will have $O(\log n)$ replanning iterations, while being more adaptive when we are closer to satisfying the total prize constraint. This is a trade-off of adaptivity vs computation time.

3. Only the *first* node is visited, after which we directly replan. This allows the algorithm to take new online information about the real prizes into account directly, but is very expensive to compute as it requires $O(n)$ iterations.

