# OpenReview forum: "Attention, Learn to Solve Routing Problems!"
_ICLR.cc/2019/Conference_

### Official Review · AnonReviewer2 · 2018-10-29
**A nice contribution to neural combinatorial optimisation**

**Rating:** 7
**Confidence:** 5

**Review:**

This paper is one of a sequence of works trying to learn heuristics for solving combinatorial optimisation problems. Compared to its predecessors, its contributions are three-fold. First, it introduces a tweak on the REINFORCE learning algorithm, outperforming more complicated methods. Second, it introduces a new model for combinatorial tasks which delivers interesting results on several tasks which are varied though related. Finally, it evaluates this model on many tasks.

****Quality and clarity****
This is a very high-quality paper.
The writing is clear and sharp, and the reading experience is quite enjoyable (the witty first paragraph sets the tone for what is to follow), even if the text is at times a bit verbose.
Another point to commend is the honesty of the paper (see e.g. the comment on the performance of the model on TSP vs specialised solvers such as Concord).
The related work section is complete and well documented.
Finally, the experimental results are clearly presented and well-illustrated.

****Originality and significance****
On the theoretical side, the contributions of this paper are interesting but not ground-breaking. The REINFORCE tweak is close to other algorithms that have been tried in the last few years (such as indeed the one presented in Rennie et al, 2016). The model architecture, while successful, is not a large departure from the Transformer presented in Vaswani et al, 2017.

More significant is the complete set of experiments on a varied subset of combinatorial tasks, which showcases one of the promises of using machine learning for combinatorial optimisation: reusability of a single model for many tasks.

****Conclusion****
Overall, this is a nice, very well-written paper. Its contributions, though not ground-breaking, are significant to the field, and constitute another step in the right direction.

Pros
- high-quality writing
- very clear
- complete experiments on a variety of tasks, some of which do not have optimal solvers
- honest assessment of the model

Cons
- the theoretical contributions are not ground-breaking (either the the tweak on REINFORCE or the model architecture)
- the model is still far from obtaining meaningful results on TSP (although it's interesting to compare to previous learned models, only solving problems with 100 nodes also illustrates how far we have to go...)

Details
- Dai et al has been published at NIPS and is no longer an arxiv preprint
- the comparison to AlphaGo should either be expanded upon or scratched. Although it could be quite interesting, as it is it's not very well motivated.

---

> ### Author Response · Authors · 2018-11-14
> **Motivation of reusing part of Transformer architecture, paper updated for detailed comments**
>
> Thank you for appreciating our paper!
>
> - Please let us motivate why we reuse a large part of the Transformer architecture. We think one of the successes of Deep Learning is the reduction of the need for manual feature engineering. We would like to avoid replacing feature engineering by task specific model engineering. Since the encoder has the generic task of learning node representations, we borrow the powerful Transformer architecture (interpreting it as an instance of a graph neural network). The decoder however is different and suitably adapted to the problem at hand.
> - Thank you for the detailed suggestions: we updated the Dai et al. reference and expanded on the AlphaGo comparison in the updated paper.

---

> > ### Comment · AnonReviewer2 · 2018-11-28
> > **thanks for your answers and the paper updates**
> >
> > I appreciate the updates to the paper. My current assessment stands.

---

### Official Review · AnonReviewer3 · 2018-11-01
**Simple application of attention + reinforce to routing problems, scalability is unclear**

**Rating:** 6
**Confidence:** 5

**Review:**

The paper presents an attention-based approach to learning a policy for solving TSP and other routing-type combinatorial optimization problems. An encoder network computes an embedding vector for each node in the input problem instance (e.g., a city in a TSP map), as well as a global embedding for the problem instance. The encoder architecture incorporates multi-head attention layers to compute the embeddings. The decoder network then uses those embeddings to output a permutation of the nodes which is used as the solution to the optimization problem. The encoder and decoder are trained using REINFORCE to maximize solution quality. Results are shown for four problem types -- TSP, vehicle routing, orienteering problem, and stochastic prize collecting TSP.

Positive aspects of the paper: The problem of learning combinatorial optimization algorithms is definitely an important one as it promises the possibility of automatically generating special purpose optimizers. Showing experimental results for different problem types is useful as it gives evidence for broad applicability. The paper is well-written, the related work section is nice, and the background material is explained well enough to make it a self-sufficient read.

I have two main criticisms:
1. Scalability of the approach: Focusing on the TSP experiments, the problem sizes of 20, 50, and 100 are really trivial for a state-of-the-art exact solver like Concorde or heuristic algorithm like LKH. And there have already been many papers showing that RL can be used for small-scale TSP and other problems (many are cited in this paper). At this point the interesting question is whether an RL approach can scale to much bigger problem instances, both in terms of solution quality as well as inference running time. For example, the DIMACS TSP Challenge problem instances have sizes up to 10^7 cities. New heuristics used with LKH (e.g. POPMUSIC) can scale to such sizes and empirically show complexity that is nearly linear with respect to the number of cities. It seems that the proposed approach would have quadratic complexity, which would not scale to much bigger problem instances. Table 2 also suggests that the solution quality (optimality gap) becomes worse for bigger sizes. If there was strong evidence that the approach could scale to much larger instances, that would have added to the novelty of the paper.

2. Insufficient comparisons:
a. The comparison to Gurobi's running time in Table 1 is misleading because in addition to outputting a solution, it also outputs a certificate of optimality. It is possible that Gurobi finds the optimal solution very quickly but then spends a large amount of time proving optimality. Since RL approaches don't prove optimality, it would be more fair to report Gurobi's time to first reach the optimal solution (and disregard proving time). This may turn out to be much smaller than the times reported in Table 1.
b. It would be good to compare against the state-of-the-art TSP-specific algorithms (Concorde, LKH) as well. Even if a general-purpose RL approach does not beat them, it would be good to assess how much worse it is compared to the best expert-designed custom algorithms so that the tradeoff between human expertise and solution quality / running time is clear.

It would also be useful to give insight into what does attention buy for the kinds of problems considered. Why do we expect attention to be helpful, and do the results match those expectations?

---

> ### Author Response · Authors · 2018-11-14
> **Comments on scalability, paper updated with additional comparisons and added intuition about attention model**
>
> Thank you for seeing the importance of the problem and the value of showing the broad applicability. Please let us address you concerns.
>
> - Scalability
> Scalability is indeed a very important direction for further research. We think that the way that heuristics (like you mention) scale almost linearly is by considering the problem locally, e.g. by local search or by limiting the set of edges for nodes (e.g. consider a sparse graph). A very promising approach would be to combine these ideas with learning, e.g. by learning how to perform local search (rather than a construction as we do here) on a sparse graph. We think our work is a step in this direction by using an architecture that could be extended to operate (potentially locally) on a (sparse) graph structure, and a powerful algorithm to train with the rollout baseline.
>
> - Insufficient comparisons
> a.
> You are right that Gurobi may spend significant time to prove optimality after finding the solution. However, we are not sure if reporting the time the solution is found as if it were the run time is the right thing to do: the algorithm cannot stop (without sacrificing performance) at this time since it has no way to know that the current solution is optimal. By the same argument, we could also report the time a solution was first found (or sampled) in a heuristic search procedure, but this is a measure 'in hindsight' which does not constitute a practical algorithm.
> Nevertheless, it's a good point that Gurobi may find good solutions early, which we can use 'heuristically' by setting a time limit or increasing the MIP gap (stop when the solution is proven within x % of optimal). We found that increasing the MIP gap (to as much as 5%) for TSP reduced running time at most 20%. A time limit of 1s makes no difference for TSP20/50 but results in no feasible solution being found in some cases for TSP100. A larger timelimit has no effect. For the OP and the PCTSP, however, we can tradeoff time for performance, but with limited success for larger instances (we added results for 1s, 10s and 30s time limit to the paper).
> b.
> We thought Concorde/LKH would not add much as Gurobi already finds optimal solutions very quickly. However, following your suggestions we ran the experiments and added the results, being that Concorde is slower for smaller instances but 6x faster for TSP100. LKH empirically finds optimal results but takes slightly longer than Gurobi.
>
> - What does attention buy
> Thank you for this suggestion (which was also noted by R1), this is indeed something that was missing which we have added to the discussion section of the paper.

---

### Official Review · AnonReviewer1 · 2018-11-03
**A good paper; missing comparison to very relevant work**

**Rating:** 7
**Confidence:** 5

**Review:**

This paper proposes an alternative deep learning model for use in combinatorial optimization. The attention model is inspired by the Transformer architecture of Vaswani et al. (2017). Given a distribution over problem instances (e.g. TSP), the REINFORCE update is used to train the attention model. Interestingly, the baseline used in the REINFORCE update is based on greedy rollout using the current model. Experimentally, four different routing problems are considered. The authors show that the proposed method often outperforms some other learning-based methods and is competitive with existing (non-learned) heuristics.

Overall, this is a good piece of work. Next, I will touch on some strengths and weaknesses which I hope the authors can address/take into account. My main concern is the lack of comparison with Deudon et al. (2018).

Strengths:
- Writing: beautifully written and precise even with respect to tiny technical details; great job!

- Versatility: the experimental evaluation on four different routing problems with different kinds of objectives and constraints, different baseline heuristics, etc., is quite impressive (irrespective of the results). The fact that the proposed model can be easily adapted to different problems is encouraging, since many real-world operational problems may be different from textbook TSP/VRP, and hard to design algorithms for; a learned algorithm can greatly expedite the process. This versatility is shared with the model in Dai et al. (2017) which applied to different graph optimization problems.

- Choice of baseline: the use of the greedy policy is definitely the right thing to do here, as one wants to beat "simpler" baselines.

- Results: the proposed method performs very well and does not seem hard to tune, in that the same model hyperparameters work well across different problems.

Weaknesses:
- Comparison to Deudon et al. (2018): I believe the authors should do more work to compare against Deudon et al. (2018). This includes expanding the sentence in related work, describing the differences in more detail; perhaps a side-by-side graphical comparison of the two models in the appendix would help; reporting results from or running the code of that paper for the relevant problems (TSP?). This is crucial, since that paper also builds on the Transformer architecture, attention, etc. Its code is also online and seems to have been up for a while (https://github.com/MichelDeudon/encode-attend-navigate). There is quite some overlap, and the reader should be able to understand how the two models/papers differ.

- Intuition: One thing that is lacking here is some intuitive explanation of *why* this particular attention model is a reasonable choice for guiding a combinatorial algorithm. For instance, earlier work such as Pointer networks or S2V-DQN each addressed certain issues with other models of the time (e.g. capturing graph structure in S2V-DQN). If the choice of the model is purely performance-driven, that is completely fine, but then it makes sense to walk the reader through the process that got you to the final model. You do some of that in the ablation study in 5.2, for the baseline. Additionally, I am wondering about why this attention model is good for a combinatorial problem.

Questions/suggestions:
- Performance metric: if I understand correctly, Table 1 reports objective values. Could you additionally report optimality gaps compared to the best solution found *across* methods (including Gurobi, especially when it solves to optimality for the smaller problems/all of TSP)? Otherwise, it is very hard to interpret the differences in absolute objective values across methods.

- Baseline: could you use a non-learned baseline (e.g. 2-opt for the case of TSP) at the beginning of the training (then go to your learned but greedy baseline)? Might this give a stronger baseline at the beginning and accelerate training?

---

> ### Author Response · Authors · 2018-11-14
> **Paper updated to include comparison to Deudon et al. (2018) and intuition about attention model**
>
> Thank you for reviewing and appreciating our paper! Please let us address your concerns; we have updated the paper according to your suggestions.
>
> - Comparison to Deudon et al.
> We would like to emphasize that Deudon et al. (2018) is *concurrent* work that actually appeared *after* we released an early version (as online preprint) of this paper. However, we agree that to the reader the comparison is relevant and we have updated the paper to include an explanation of the differences. Also, since results in their paper are not directly comparable, we ran their code with the same number of iterations and samples as we do (this improved the results). We have added the numbers in Table 1.
>
> - Intuition
> Thank you for this helpful suggestion (which was also mentioned by R3). Focusing on the technical parts of the paper and the results this is indeed something that we overlooked. We have added this to the discussion section.
>
> - Results as percentages
> Not reporting the percentages was merely a practical matter to keep things clear and save space. We have added the percentages in the updated paper and allocated more space for the table to keep things as clear as possible.
>
> - Baseline
> Indeed, using a known algorithm as baseline is a good suggestion. This is an interesting direction for future work.

---

> > ### Comment · AnonReviewer1 · 2018-12-04
> > **Great job**
> >
> > Wonderful - I have updated my score accordingly.

---

### Public Comment · (anonymous) · 2023-05-06
**Experiments May Have Some Issue**

Hi,

I really enjoy your excellent work! But I have found some issue in your implementation.

You set the parameter MAX_TRAILS to 10000 in your implementation of the baseline LKH3. I personally set it to 20 (which is the default set of LKH3 running on a 20-nodes CVRP) and get the same optimal results while the running time can be 200x faster. So I question whether the comparison is fair enough.

I would really appreciate it if someone can answer my doubt.

---

> ### Author Response · Authors · 2023-05-30
> **You are right but doesn't affect main conclusions**
>
> Hi,
>
> Thanks a lot for your feedback as you raise a fair point. I am not sure why I set MAX_TRIALS to 10000 (it has been some years), probably this was to improve performance for the 100 node instances.
>
> In any case, I think the 20 node experiments are not really representative: it is trivial to solve TSP (and also VRP) with just 20 nodes to optimality in very short time, so the experiments are mostly there to compare to previous works and show the gap to optimality. I fully acknowledge that LKH is better and I'm sorry if the paper raises a different impression. I think the conclusion is much more fair for 100 node instances (where still LKH is better).
>
> In general it is hard to do a general and fair comparison taking into account hardware differences (CPU/GPU) and tradeoffs (time vs running time) which is acknowledged in the paper. In hindsight, I would do things differently and indeed in a follow up paper, we did a better job by reporting HGS (as being better than LKH) as well and reporting the quality as a function of runtime (see https://arxiv.org/pdf/2102.11756.pdf).
>
> In all, I think your concern is valid, but I also think they don't affect the main conclusions and value of the paper: we show that it is possible to learn constructive policies which are able to construct good solutions quickly while learning from scratch for different vehicle routing problems.
>
> My compliments for your observations, I think it is great that you point this out as we should keep each other to high standards and fair evaluation of algorithms. Please let me know if you have further questions/concerns.

---

### Meta-Review · Area_Chair1 · 2018-12-14

**Confidence:** 4
**Recommendation:** Accept (Poster)

**Metareview:**

The paper presents a new deep learning approach for combinatorial optimization
problems based on the Transformer architecture. The paper is well written
and several experiments are provided. A reviewer asked for more intuition to
the proposed approach and authors have responded accordingly. Reviewers are
also concerned with scalability and theoretical basis.
Overall, all reviewers were positives in their scores, and I recommend accepting the paper.